# Spatiotemporal patterns of neocortical activity around hippocampal sharp-wave ripples

**J Karimi Abadchi[1], Mojtaba Nazari-Ahangarkolaee[1], Sandra Gattas[2,3], Edgar Bermudez-Contreras[1], Artur Luczak[1], Bruce L McNaughton[1,4]\*, Majid H Mohajerani[1]\***

[1]Canadian Centre for Behavioral Neuroscience, University of Lethbridge, Lethbridge, Canada; [2]Department of Electrical Engineering and Computer Science, University of California, Irvine, United States; [3]Medical Scientist Training Program, University of California, Irvine, United States; [4]Department of Neurobiology and Behavior, University of California, Irvine, United States

**Abstract** A prevalent model is that sharp-wave ripples (SWR) arise 'spontaneously' in CA3 and propagate recent memory traces outward to the neocortex to facilitate memory consolidation there. Using voltage and extracellular glutamate transient recording over widespread regions of mice dorsal neocortex in relation to CA1 multiunit activity (MUA) and SWR, we find that the largest SWR-related modulation occurs in retrosplenial cortex; however, contrary to the unidirectional hypothesis, neocortical activation exhibited a continuum of activation timings relative to SWRs, varying from leading to lagging. Thus, contrary to the model in which SWRs arise 'spontaneously' in the hippocampus, neocortical activation often precedes SWRs and may thus constitute a trigger event in which neocortical information seeds associative reactivation of hippocampal 'indices'. This timing continuum is consistent with a dynamics in which older, more consolidated memories may in fact initiate the hippocampal-neocortical dialog, whereas reactivation of newer memories may be initiated predominantly in the hippocampus.

**\*For correspondence:**
bruce.mcnaughton@uleth.ca (BLMN);
mohajerani@uleth.ca (MHM)

**Competing interests:** The authors declare that no competing interests exist.

## Introduction

Beginning with the theoretical work of *Marr (1971)*, the idea that hippocampal-neocortical interactions during slow-wave sleep (SWS) play an important role in the process of systems memory consolidation has become a dominant paradigm in memory research (*Buzsáki, 1989*; *McClelland et al., 1995*; *Wilson and McNaughton, 1994*). This idea is supported by observations of replay of recently active neural ensemble patterns in hippocampus (*Kudrimoti et al., 1999*; *Nádasdy et al., 1999*; *Pavlides and Winson, 1989*; *Skaggs and McNaughton, 1996*; *Wilson and McNaughton, 1994*) and neocortex (*Euston et al., 2007*; *Hoffman and McNaughton, 2002*; *Ji and Wilson, 2007*; *Jiang et al., 2017*; *Qin et al., 1997*), a key role of SWS in enabling structural rearrangements of neocortical synaptic connections (*Yang et al., 2014*), and effects of interruptions of SWS on memory (*Gais et al., 2007*). More specifically, replay of recent hippocampal patterns has been shown to be concentrated in SWR bursts (transient, 100–250 Hz, local field potential oscillations) and that SWRs significantly predict reactivation of neocortical ensembles during upstates (*Peyrache et al., 2009*). Finally, post-training, closed-loop interruption of SWR has a negative effect on memory for tasks that normally depend on the hippocampus for acquisition (*Ego-Stengel and Wilson, 2010*; *Girardeau et al., 2009*). Altogether, these works suggest that interactions between neocortex and hippocampus during SWS support memory processes.

A prevalent model is that SWRs arise 'spontaneously' in CA3, which could lead to memory retrieval due to attractor network dynamic, and propagate outward to the neocortex through CA1 to enable memory consolidation there (*Buzsáki, 1989*; *Csicsvari et al., 2000*; *Shen and McNaughton, 1996*). Moreover, the hippocampal memory indexing theory (HMIT) and the complementary learning systems theory (CLST) are two other prominent theories pointing to a functional relationship between neocortex and hippocampus for the retrieval of memories and their integration into neocortical connections. According to HMIT (*McNaughton, 2010*; *Teyler and DiScenna, 1986*), the hippocampus generates an 'index code' which is stored within weakly interacting neocortical processing modules at the time of the experience and serves to coordinate retrieval of complete memories either during behavior or sleep. According to the CLST, repetition of this reactivation process is believed to enable a consolidation process whereby the stored information is refined and eventually encoded by intra-neocortical connections in a form that is more categorically structured (*McClelland et al., 1995*; *Winocur and Moscovitch, 2011*) and relatively independent of the hippocampus. The increasing evidence favoring this memory-consolidation-by-replay theory suggests that a more detailed study of the spatiotemporal relationships between SWR and neocortical activation during sleep is of paramount importance.

Previous studies, using primarily electrophysiological methods, have focused on functional relationships between SWRs and a single or a few discrete neocortical regions (*Alexander et al., 2018*; *Battaglia et al., 2004*; *Khodagholy et al., 2017*; *Mölle et al., 2006*; *Opalka et al., 2020*; *Siapas and Wilson, 1998*; *Sirota et al., 2003*). These studies have shown that a given neocortical region can activate and deactivate around SWR times, which reflects the sleep-related neocortical slow oscillation (SO), and that SO can be spatially local or global propagating waves (*Massimini et al., 2004*; *Mohajerani et al., 2010*). Moreover, recently combined fast time-scale electrophysiological techniques with whole brain Blood Oxygen Level-Dependent (BOLD) fMRI imaging provided a snapshot of the cooperative patterns of the large numbers of brain structures involved either leading to or responding to hippocampal SWRs (*Logothetis et al., 2012*; *Ramirez-Villegas et al., 2015*). Although these studies have improved our understanding of hippocampal-neocortical interactions, their conclusions were limited because: (a) the electrophysiological approaches have been unable to resolve the regional structure and dynamics of HPC-NC interaction across a large neocortical area due to relatively sparse spatial sampling (b) fMRI studies generally rely on an indirect measures of neuronal activity and the nature of the underlying neuronal activity correlate (i.e. synaptic and spiking activity) remain unclear. Given the physiological speed of neuronal communication, the cause-effect relationships are hard to decipher with certainty with the resolution at the scale of seconds of the fMRI method. In the present study we expand upon previous theoretical and experimental studies. We combined wide-field optical imaging of dorsal neocortex of mice, covering most of sensory and motor cortices and some association areas, with concurrent electrophysiological monitoring of hippocampal SWRs and MUA during natural sleep and under urethane anesthesia, which produces a state that is in many respects similar to natural sleep (*Barthó et al., 2014*; *Clement et al., 2008*; *Pagliardini et al., 2013b*).

We report both a spatial and temporal continuum in the degree and onset timing of modulation of neocortical activity around SWRs, thereby providing a spatiotemporal map of potential interactions between hippocampus and neocortex.

## Results

### Experimental protocol for investigating dynamics of hippocampal-neocortical interactions during sleep

Using in vivo wide-field of view mesoscale optical imaging and voltage-sensitive dye (VSD) (*Bermudez-Contreras et al., 2018*; *Ferezou et al., 2007*; *Kyweriga et al., 2017*; *Mohajerani et al., 2013*; *Shoham et al., 1999*) and genetically encoded sensor of extracellular glutamate (iGluSnFR) (*Marvin et al., 2013*; *Xie et al., 2016*; *Figure 1—figure supplement 1*), we imaged neocortical activity dynamics from a large cranial window in the right hemisphere and combined it with electrophysiology in the ipsilateral hippocampus (*Figure 1A–C*; *Figure 1—figure supplement 1*; *Figure 1—video 1*) to capture hippocampal SWRs and MUA. Unlike calcium reporters (e.g., GCaMP6) with slow kinetics, both VSD and iGluSnFR measures electrical activity with relatively

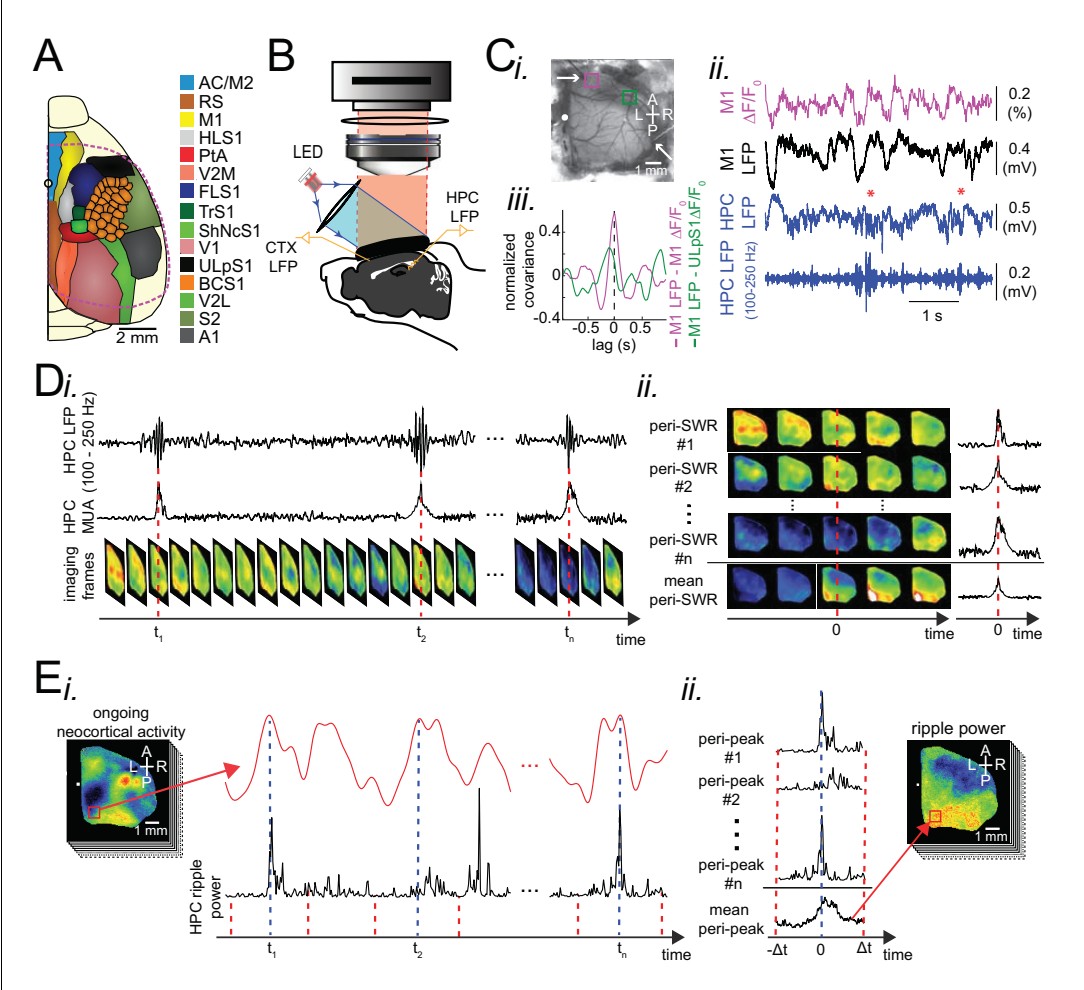

**Figure 1.** Experimental protocol for investigating dynamics of neocortical-hippocampal interactions during sleep. (**A**) Schematic of a cranial window for wide-field optical imaging of neocortical activity using voltage or glutamate probes. The voltage or glutamate signal was recorded from dorsal surface of the right neocortical hemisphere, containing the specified regions. The red dashed line marks the boundary of a typical cranial window. The abbreviations denote the following cortices AC/M2: anterior cingulate/secondary Motor, RS: retrosplenial, M1: primary motor, HLS1: hindlimb primary somatosensory, PtA: posterior parietal, V2M: secondary medial visual, FLS1: forelimb primary somatosensory, TrS1: trunk primary somatosensory, ShNcS1: shoulder/neck primary somatosensory, V1: primary visual, ULpS1: lip primary somatosensory, BCS1: primary barrel, V2L: secondary lateral visual, S2: secondary somatosensory, A1: primary auditory. (**B**) Schematic of the experimental setup for simultaneous electrophysiology and wide-field optical imaging. A CCD camera detects reflected light coming from fluorescent indicators, in the superficial neocortical layers, which are excited by the red or blue LEDs. An additional infra-red camera recorded pupil diameter (not shown). Hippocampal LFP recordings were conducted for SWR and MUA detection. A neocortical LFP recording was also acquired to compare imaging and electrophysiological signals. (**C**) (i) Photomicrograph of the wide unilateral craniotomy with bregma indicated by a white circle in each image. Compass arrows indicate anterior (A), posterior (P), medial (M) and lateral (L) directions. The white arrow indicates a neocortical LFP electrode position. (ii) Exemplar voltage signal recorded from a region in the M1 indicated by a magenta square in Ci, aligned with neocortical and hippocampal LFPs, and a hippocampal trace filtered in the ripple band. Asterisks indicate detected SWRs. (**D**) Schematic figure demonstrating the peri-SWR averaging of neocortical activity. (i) Ripple band filtered LFP trace displaying three example SWRs (top row) and hippocampal multi-unit activity (MUA) trace (middle row), temporally aligned with concurrently recorded neocortical voltage activity (bottom row). (ii) For each detected SWR, corresponding neocortical imaging frames (left rows) and MUA traces (right rows) are aligned with respect to SWR centers and then averaged (bottom rows). (**E**) Demonstration of how peri-neocortical-peak-activation average ripple power was calculated. (i) The red trace is the voltage signal from the indicated region of interest (red square) shown in the image on the left. The black trace is the temporally aligned hippocampal ripple power time series. Blue dashed lines are the timestamps of three detected peak activations in the indicated neocortical region. (ii) For each detected peak activation, ripple power traces were aligned and averaged. This figure has two figure supplements.

The online version of this article includes the following video and figure supplement(s) for figure 1:

**Figure supplement 1.** Expression level of iGluSnFR and electrode localization.

**Figure supplement 2.** Physiological signatures of different brain states in the head-restrained mouse.

**Figure 1—video 1.** Spontaneous neocortical voltage activity with concurrently recorded hippocampal LFP and ripple activity.

*Figure 1 continued on next page*

*Figure 1 continued*

https://elifesciences.org/articles/51972#fig1video1

**Figure 1—video 2.** Wake/sleep scoring.

https://elifesciences.org/articles/51972#fig1video2

---

high temporal resolution (*Xie et al., 2016*). Due to the high level of synchrony in neocortical activity (*Mohajerani et al., 2010*) and hippocampal SWR activity (*Chrobak and Buzsáki, 1996*) across hemispheres, we decided to record unilaterally from neocortex and hippocampus. Imaging unilaterally has the advantage of larger coverage of neocortical regions, particularly more lateral ones like primary auditory cortex. In some of the conducted VSD imaging experiments, a bipolar electrode was utilized to record neocortical LFP. The wide-field optical imaging of membrane potential activity in a given region of interest was correlated with LFP activity recorded from the same neocortical site confirming that the optical voltage signal reflects underlying electrophysiological processes (*Arieli et al., 1995*; *Figure 1C*). We conducted our experiments under both natural sleep and urethane anesthesia. Urethane anesthesia has been reported to model natural sleep; In particular, characteristic electrophysiological signatures of slow-wave sleep (SWS) including up- and down-states, delta waves, spindles, and hippocampal SWR are present under urethane anesthesia (*Barthó et al., 2014*; *Clement et al., 2008*; *Pagliardini et al., 2013a*; *Wolansky et al., 2006*; *Figure 1Cii*; *Figure 1—video 1*). We confirmed that the quantities, reported in the following sections, were not different across urethane anesthesia versus sleep and VSD versus iGluSnFR imaging conditions (*Figure 2—figure supplement 3*; *Figure 3—figure supplement 1*; *Figure 4—figure supplement 2*; *Figure 5—figure supplement 2*; *Figure 6—figure supplement 2*; Figure 2—figure supplement 7). For natural sleep experiments, conducted on iGluSnFR mice under head-restrained condition, we used neck muscle electromyogram (EMG) recordings and hippocampal delta-to-theta band power ratio to score sleep state. Sleep scores were further validated by monitoring pupil diameter from each animal and comparing the distribution of physiological measures in different sleep states (*Figure 1—figure supplement 2*; *Figure 1—video 2*). During the natural sleep experiments, pupil dilation measurement was possible since mice slept with eyelids partially open (*Yüzgeç et al., 2018*).

## Patterns of activity in neocortical regions are differentially modulated around hippocampal SWRs

We began by investigating how different neocortical regions activate and deactivate around SWRs times. To do so, we performed SWR-triggered averaging of neocortical activity, (*Figure 1D*; *Figure 2A–B*; *Figure 2—figure supplement 1*; *Figure 2—videos 1*, *2* and *3*).~250 ms before the center of SWRs (t = 0; see Materials and methods), most neocortical regions transiently deactivated. Deactivation was followed by a strong activation in most regions in close temporal proximity to SWRs (*Figure 2A–B*). Since these phenomena occurred in both head-restrained sleep and under urethane anesthesia, and the data exhibited a high degree of similarity (*Figure 2—figure supplements 2* and *3*), we pooled the urethane and head-restrained sleep data for group analysis. Out of all imaged regions, RSC was the most strongly modulated, showing the highest peak of activation (Figure 3—figure supplement 3Ci) and second highest peak of deactivation (*Figure 2—figure supplement 3Cii*). Interestingly, regions with comparable magnitudes of activations and deactivations tended to fall within previously identified neocortical structural subnetworks (*Zingg et al., 2014*; *Figure 2Ci*). The medial subnetwork that includes RSC and posterior parietal cortex (PtA) along with the visual subnetwork showed the highest degrees of peri-SWR activation (*Figure 2Cii–iii*; n = 14 per subnetwrok; repeated measure ANOVA with Greenhouse-Geisser correction for sphericity: $F_{3,39}$ = 44.303, p=2.2193 $\times$ $10^{-10}$; post-hoc multiple comparison with Tuckey's correction: medial vs visual p=0.11844, medial vs auditory p=7.8147 $\times$ $10^{-5}$, medial vs somatomotor p=1.1128 $\times$ $10^{-6}$, visual vs auditory p=0.00024702, visual vs somatomotor p=2.9247 $\times$ $10^{-05}$, auditory vs somatomotor p=0.19802) and deactivation (*Figure 2D*; n = 14 per subnetwrok;repeated measure ANOVA with Greenhouse-Geisser correction for sphericity: $F_{3,39}$ = 15.508, p=1.1009 $\times$ $10^{-5}$; post-hoc multiple comparison with Tuckey's correction: medial vs visual p=0.63021, medial vs auditory p=0.014438, medial vs somatomotor p=0.00076384, visual vs auditory p=0.0042707, visual vs somatomotor

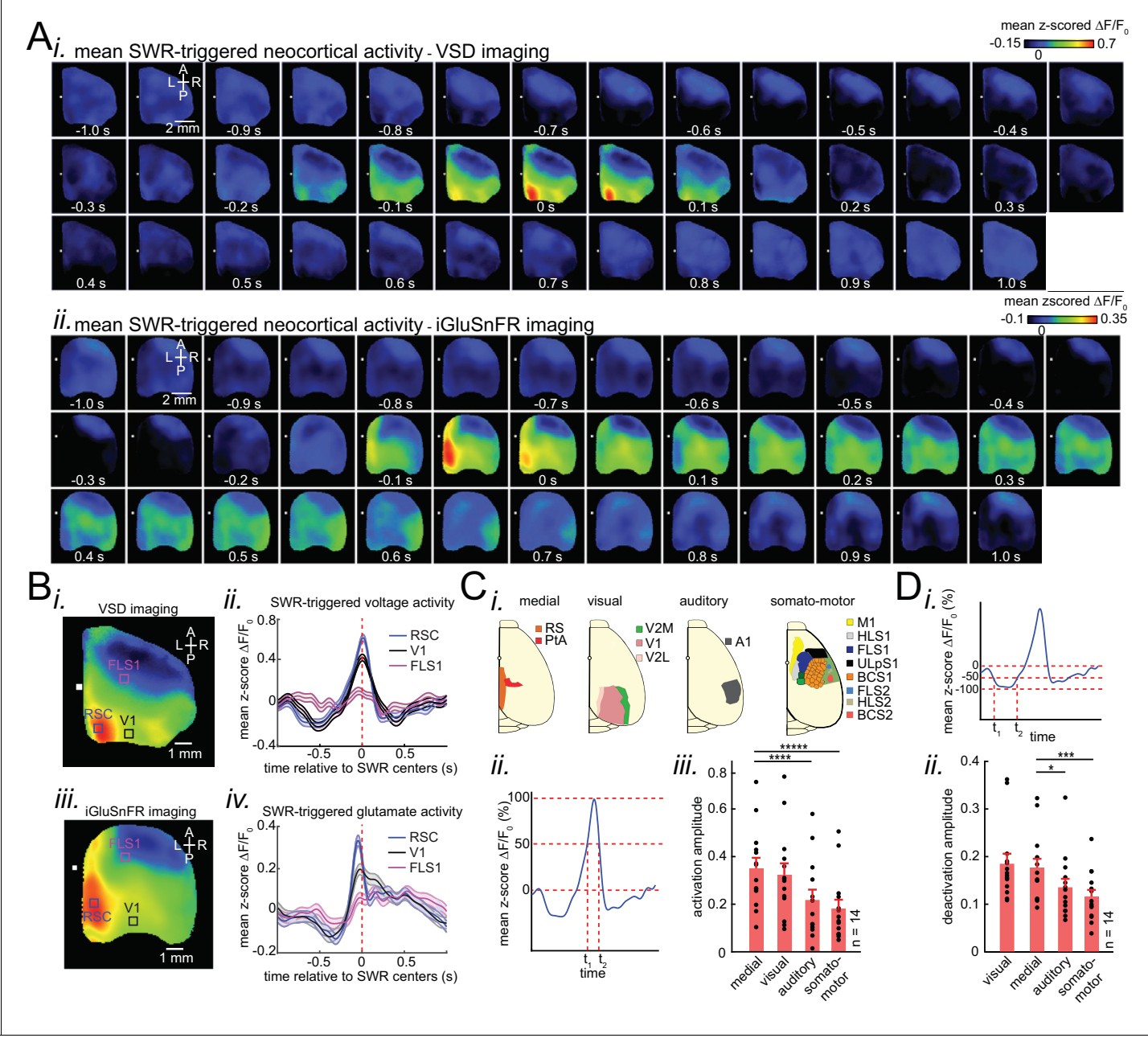

**Figure 2.** Patterns of activity in neocortical regions are differentially modulated around hippocampal SWRs. (**A**) Representative montage of mean peri-SWR neocortical activity measured using (i) voltage-sensitive dye and (ii) glutamate-sensing fluorescent reporter iGluSnFR under urethane anesthesia and head-restrained natural sleep, respectively. 0s-time indicates SWR centers. Images have been z-scored and scaled to the depicted color bars. (**B**) (i–iv) Example traces showing voltage or iGluSnFR signals from selected regions in (i) and (iii). Plots are the average of optical signals measured from $3 \times 3$ pixel boxes (~0.04 mm²) placed within retrosplenial (blue), visual (black), and forelimb somatosensory (magenta) cortices. The thickness of the shading around each plot indicates SEM. 0 s-time indicates SWR centers. (**C**) (i) Four major structurally defined neocortical subnetworks (medial, visual, auditory and somato-motor). (ii) Demonstration of how the activation amplitudes were quantified. The activation amplitude was defined as the mean of the signal across full-width at half maximum ($t_1$ to $t_2$). (iii) Grand average (n = 14 animals) of activation amplitudes across neocortical subnetworks, sorted in decreasing order. Each data point is the average of activation amplitudes of all regions in a given subnetwork and in a given animal. (repeated measure ANOVA with Greenhouse-Geisser correction for sphericity: $F_{3,39} = 44.303$, p=$2.2193 \times 10^{-10}$; post-hoc multiple comparison with Tuckey's correction: medial vs visual p=0.11844, medial vs auditory p=$7.8147 \times 10^{-5}$, medial vs somatomotor p=$1.1128 \times 10^{-6}$, visual vs auditory p=0.00024702, visual vs somatomotor p=$2.9247 \times 10^{-05}$, auditory vs somatomotor p=0.19802) (**D**) (i–ii) Same measurements as in C, but for neocortical deactivations preceding SWRs. Note that the deactivation peaks were rectified for group comparison in (ii). A higher value of deactivation amplitude indicates stronger deactivation. Bar graphs indicate mean ± SEM. (repeated measure ANOVA with Greenhouse-Geisser correction for sphericity: F3,39 = 15.508,

*Figure 2 continued on next page*

*Figure 2 continued*

p=1.1009 × 10–5; post-hoc multiple comparison with Tuckey's correction: medial vs visual p=0.63021, medial vs auditory p=0.014438, medial vs somatomotor p=0.00076384, visual vs auditory p=0.0042707, visual vs somatomotor p=0.0014657, auditory vs somatomotor p=0.5317). This figure has five figure supplements.

The online version of this article includes the following video, source data, and figure supplement(s) for figure 2:

**Source data 1.** Differential activation amplitude across neocortical subnetworks around SWRs.
**Source data 2.** Differential deactivation amplitude across neocortical subnetworks around SWRs.
**Figure supplement 1.** Representative spatiotemporal patters of neocortical activity around hippocampal SWRs.
**Figure supplement 2.** Characteristics of SWRs recorded in urethane anesthesia, head-restrained and unrestrained naturally sleeping mice.
**Figure supplement 3.** Differential modulation of neocortical regions under sleep/urethane anesthesia and VSD/iGluSnFR imaging conditions are similar.
**Figure supplement 4.** Differential modulation of neocortical regions does not depend on SWR detection threshold value.
**Figure supplement 5.** Differential modulation of neocortical activity, measured by LFP, around SWRs.
**Figure 2—video 1.** Mean peri-SWR voltage activity under urethane anesthesia.
https://elifesciences.org/articles/51972#fig2video1
**Figure 2—video 2.** Mean peri-SWR iGluSnFR activity during head-restrained sleep anesthesia.
https://elifesciences.org/articles/51972#fig2video2
**Figure 2—video 3.** Mean peri-SWR iGluSnFR activity under urethane anesthesia.
https://elifesciences.org/articles/51972#fig2video3

p=0.0014657, auditory vs somatomotor p=0.5317), respectively, followed by auditory and somatomotor networks. Similar results were obtained using higher thresholds for detecting SWRs (*Figure 2—figure supplement 4*). In addition, we replicated some of these results using electrophysiology in unrestrained naturally sleeping mice in another cohort of animals (*Figure 2—figure supplement 5*).

## Ripple power is distinctively correlated with peak activity in different neocortical subnetworks

Having observed that neocortical regions are strongly modulated around SWRs, we investigated whether the reverse is also the case, that is Does ripple power increase when strong activations take place in a given neocortical region? To address this question, the ripple power traces centered on peak activations (see Methods) in each neocortical pixel (or region of interest) were averaged and compared across neocortical subnetworks (*Figure 1E*). Ripple power indeed increased relative to peak neocortical activations, predominantly in regions that were most modulated around SWRs (*Figure 3A–B*). Ripple power increased most around peak activations in the visual and medial compared with auditory and somatomotor subnetworks (*Figure 3C*; n = 14 per subnetwrok; repeated measure ANOVA with Greenhouse-Geisser correction for sphericity: $F_{3,39}$ = 14.172, p=1.8183 × $10^{-5}$; post-hoc multiple comparison with Tuckey's correction: medial vs visual p=0.99476, medial vs auditory p=0.073032 [p=0.0152 without Greenhouse-Geisser correction], medial vs somatomotor p=0.00054029, visual vs auditory p=0.030248, visual vs somatomotor p=0.0010466, auditory vs somatomotor p=0.17794). Thus, during natural sleep and under urethane anesthesia (*Figure 3—figure supplement 1*), the medial and visual subnetworks are more likely to be coordinated with hippocampal SWRs. Moreover, we investigated the relationship between neocortical slow oscillation and hippocampal SWRs in animals for which neocortical electrophysiological signals were recorded (*Figure 3—figure supplement 2*). We found that hippocampal ripple power is suppressed (elevated) before (after) the onset of neocortical up-states. Similarly, hippocampal ripple power is elevated (suppressed) before (after) the onset of neocortical down-states. Therefore, the results obtained using optical imaging and neocortical electrophysiology were compatible.

## Neocortex tends to activate sequentially from medial to more lateral regions around SWRs

After identifying which neocortical subnetworks are modulated around SWRs, we investigated the temporal order in which neocortical regions activated. The timestamp of the peak activity ($t_p$ in *Figure 4Ai*) in the mean peri-SWR traces for each neocortical region was detected on an individual animal basis. We then sorted peak activation times from earliest to latest relative to SWRs centers. A

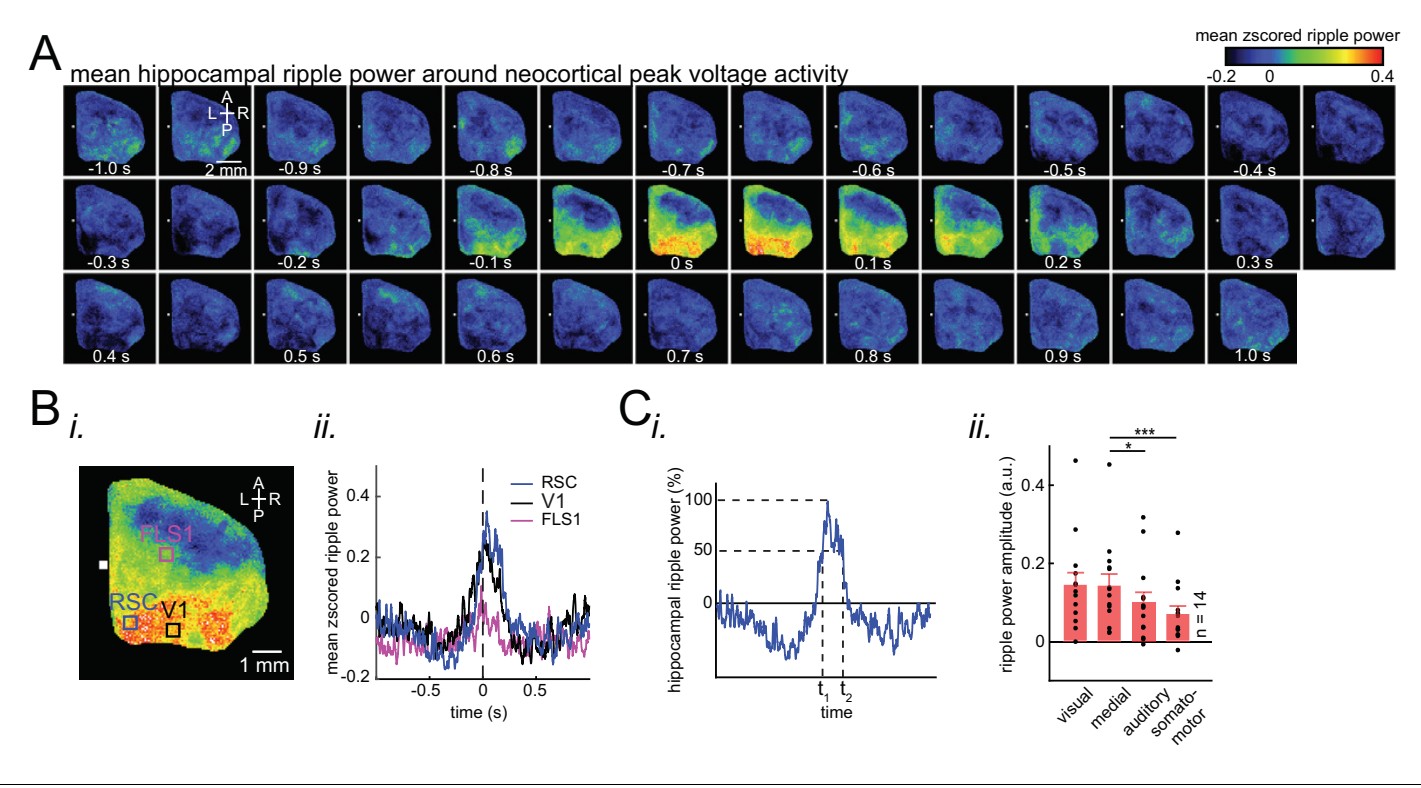

**Figure 3.** Ripple power is distinctively correlated with peak activity in different neocortical subnetworks. (**A**) Representative montage (50 ms intervals) showing spatiotemporal pattern of mean hippocampal ripple power fluctuations around the peak of neocortical activations. 0s-time indicates peak activation time in each neocortical pixel. Color bar represents the mean z-scored ripple power associated with peak activation in a given neocortical pixel. (**B**) (i–ii) A representative frame showing the spatial distribution of ripple power at neocortical peak activation time. Colored squares represent three regions of interest (retrosplenial, visual, and forelimb somatosensory cortices in blue, black and magenta, respectively) for which their associated ripple power traces are displayed in (ii). (**C**) (i) Illustration of how the ripple power amplitude was quantified. Ripple power amplitude was defined as the mean ripple power signal across full-width at half maximum ($t_1$ to $t_2$). (ii) Grand average (n = 14 animals) of mean hippocampal ripple power amplitudes across neocortical subnetworks shown in *Figure 2Ci*, sorted in decreasing order. Each data point is the average of hippocampal ripple power amplitudes associated with all the regions in a given subnetwork and in a given animal. Bar graphs indicate mean ± SEM. (repeated measure ANOVA with Greenhouse-Geisser correction for sphericity: $F_{3,39}$ = 14.172, p=1.8183 × 10–5; post-hoc multiple comparison with Tuckey's correction: medial vs visual p=0.99476, medial vs auditory p=0.073032 [p=0.0152 without Greenhouse-Geisser correction], medial vs somatomotor p=0.00054029, visual vs auditory p=0.030248, visual vs somatomotor p=0.0010466, auditory vs somatomotor p=0.17794). This figure has two figure supplements.
The online version of this article includes the following source data and figure supplement(s) for figure 3:

**Source data 1.** Correspondence of ripple power with peak activity in different neocortical subnetworks.
**Figure supplement 1.** The correlation of ripple power with neocortical peak activity is similar under sleep/urethane anesthesia and VSD/iGluSnFR imaging conditions.
**Figure supplement 2.** Modulation of hippocampal ripple power around neocortical up-/down-states.

medial to lateral temporal gradient in peak times was observed across neocortical regions (*Figure 4Aii*; repeated measure ANOVA: $F_{8,104}$ = 8.357195, p<0.0001; post-hoc test for linear trend: slope = 0.003252024 s/region, p<0.0001). The group average of peak times for each region across all animals also supported the medio-lateral direction of activations (*Figure 4Aiii*; *Figure 4—figure supplement 1*; *Figure 4—figure supplement 2*).

To investigate whether significant directional order of activation can be observed on an individual SWR basis, rather than being driven by the mean across all SWRs, we iteratively applied optical flow analysis (*Afrashteh et al., 2017*) on mean peri-SWR neocortical activity around smaller subsets (n ~ 20) of randomly chosen SWRs and calculated the direction of propagation of activity in each region (blue circular distribution in *Figure 4B* and *Figure 4—figure supplement 1C*). We observed that around SWRs, the waves of activity in medial neocortical regions showed the strongest medio-lateral directional component, and the strength of this directionality decreased in more lateral

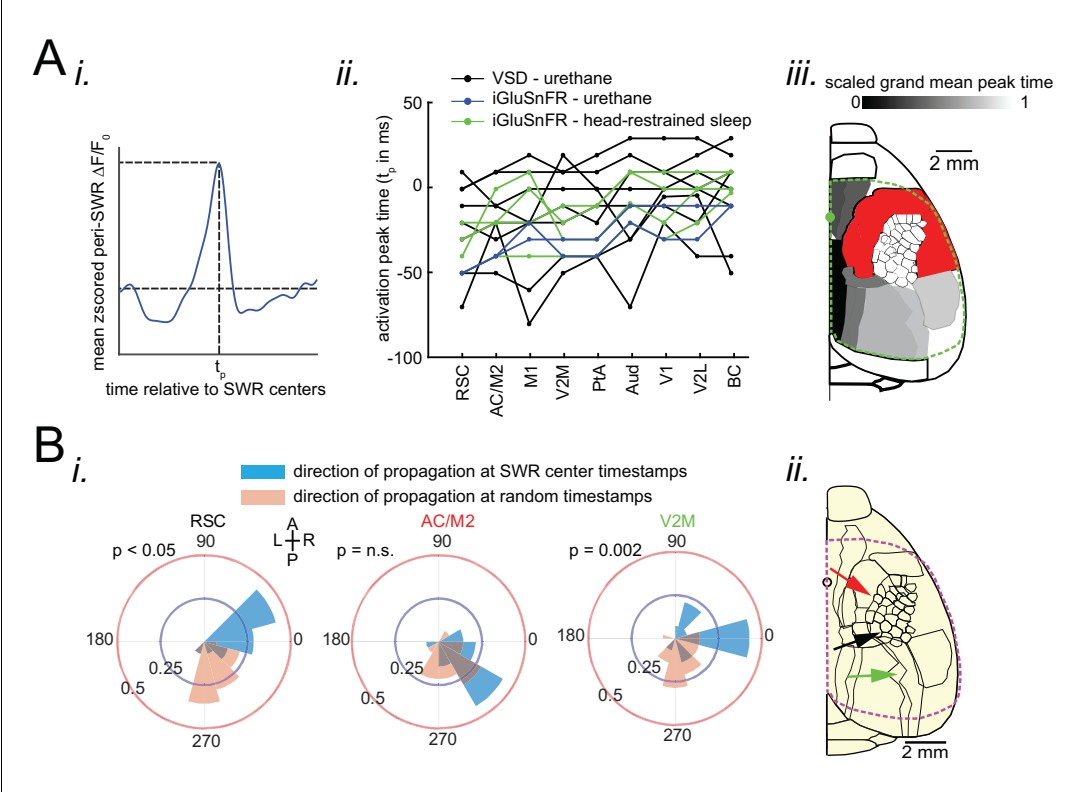

**Figure 4.** Neocortex tends to sequentially activate from medial to more lateral regions around SWRs. (**A**) (i) Demonstration of how peri-SWR neocortical activation peak time ($t_p$) was quantified. The mean peri-SWR neocortical activity trace was generated for each region (blue trace) and the timestamp of the peak was defined as $t_p$. (ii) Peri-SWR activation peak timestamp ($t_p$) relative to SWR centers (0s-time) across neocortical regions sorted in ascending order. Each line graph represents one animal. $t_p$ values were not detected in some regions and in some animals (three data points in total), mainly because there was not a strong activation in those regions. Such missing data points were filled by average of available data points in the same region and in other animals (repeated measure ANOVA: F8,104 = 8.357195, p<0.0001; post-hoc test for linear trend: slope = 0.003252024 s/region, p<0.0001). (iii) Spatial map of peri-SWR activation peak time across all animals (n = 14) indicating a medial-to-lateral direction of activation. The red area was not included in this analysis because it was not activated strong enough to yield a reliable result. (**B**) (i) Circular distributions represent the direction of propagating waves of activity in three distinct neocortical regions at hippocampal SWR (blue distribution) and at random timestamps generated by shuffling inter-SWR time intervals (red distribution). 180–0 and 90–270 degrees represent the medio-lateral and antero-posterior axes, respectively. P-values come from Kuiper two-sample test. (ii) Schematic of propagation directions measured at SWR timestamps in three neocortical regions located in the medial neocortex. This figure has two figure supplement.

The online version of this article includes the following figure supplement(s) for figure 4:

**Figure supplement 1.** Neocortex tends to activate sequentially from medial to more lateral regions around SWRs.

**Figure supplement 2.** The order of sequential activation across neocortical regions around SWRs is similar under sleep/urethane anesthesia and VSD/iGluSnFR imaging conditions.

regions (*Figure 4B*; *Figure 4—figure supplement 1*). In order to verify that the directionalities observed in optical flow analysis were meaningful; we compared them to directionalities in the same region but during random timestamps generated by shuffling the inter-SWR time intervals (orange circular distribution in *Figure 4B*; *Figure 4—figure supplement 1C*) and found a statistical difference between the two circular distributions in several regions (*Figure 4B*; Kuiper two-sample test; RSC p<0.05; AC/M2 p>0.05; V2M p=0.002; *Figure 4—figure supplement 1C*).

## Neocortical activation latency relative to SWRs spans a wide spectrum of negative to positive values

On average, neocortical voltage and glutamate activity tended to peak before SWRs in almost all of the imaged regions (*Figure 4Aii*). To investigate this temporal relationship further, we focused on RSC as an exemplar region, since it was the most modulated region around SWRs. Investigation on an individual SWR basis showed that the timings of RSC peak activity relative to SWRs forms a

continuum spanning from negative (before) to positive (after) values (**Figure 5—figure supplement 1A**). To clarify what properties of SWRs co-vary with the temporal order of RSC activation relative to SWRs, we calculated an Asymmetry Index (AI) (**Figure 5A–B**) which is defined as the difference between mean activity in the intervals $\pm \Delta t$ after and before the SWRs center, divided by their sum, such that positive and negative values represent activity tending to follow or to precede the SWR, respectively. AI was used instead of peak time because, AI, which is calculated based on integral of a signal, is less noisy, and hence more reliable (**Figure 5—figure supplement 1B**). AIs were calculated for peri-SWR RSC traces, and are referred as RSC AI (**Figure 5A–B**). As expected, RSC AI values were strongly correlated with AI values similarly calculated for regions in medial and visual subnetworks (PtA = 0.78 ± 0.03; AC/M2 = 0.78 ± 0.02; V2M = 0.897 ± 0.01; V1 = 0.67 ± 0.02; V2L = 0.68 ± 0.02; Aud = 0.51 ± 0.03; M1 = 0.46 ± 0.04; BC = 0.53 ± 0.02; FLS1 = 0.38 ± 0.03; ULpS1 = 0.23 ± 0.04). We then partitioned the distribution of RSC AI values into lower and higher quartile ranges (QR) according to whether the RSC AI range fell below the first quartile (RSC QR1)

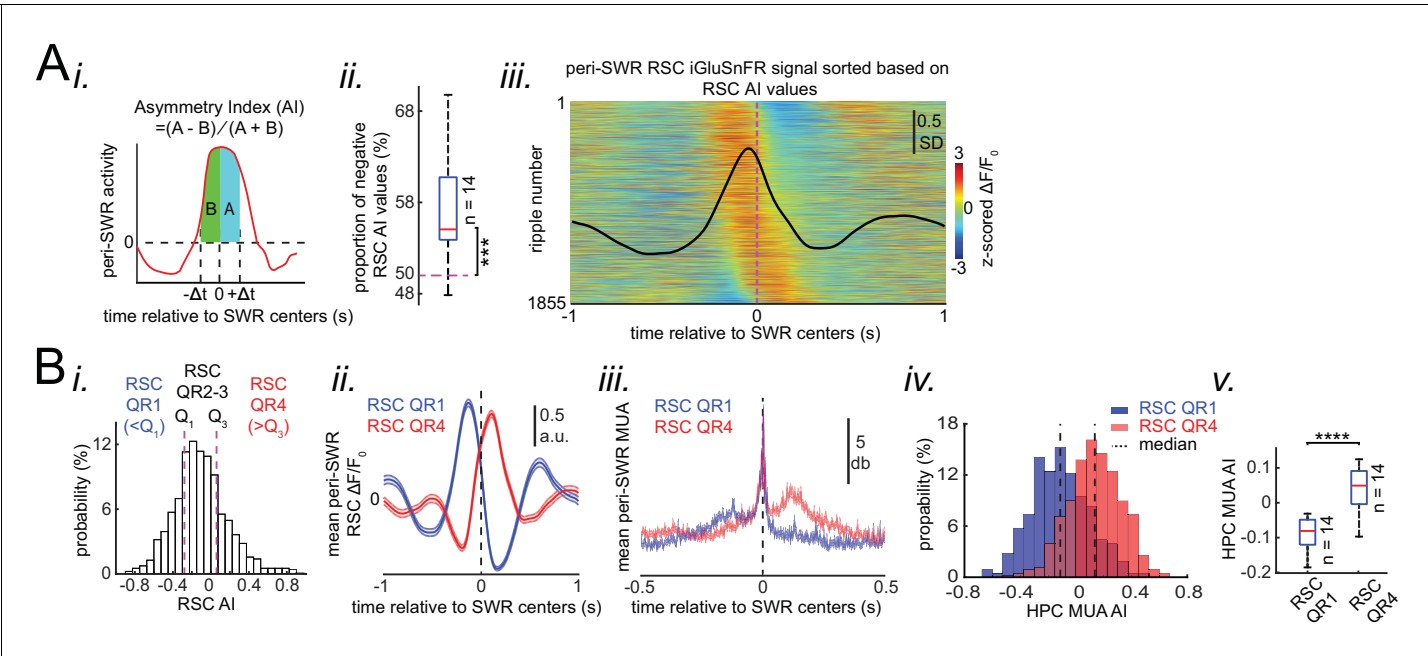

**Figure 5.** Neocortical activation latency relative to SWRs spans a wide spectrum of negative to positive values. (**A**) (**i**) Schematic of Asymmetry Index (AI) calculation. In this figure, AI was calculated for individual peri-SWR retrosplenial cortex (RSC) traces and called RSC AI. RSC AI values were used to quantify the latency of neocortical activation relative to SWR timestamps. (**ii**) The proportion of negative RSC AI values across animals (n = 14). Magenta dashed line indicates the chance level (50%). 55% (median, indicated by a red line) of peri-SWR RSC activity across animals have negative AI, meaning that on average, neocortical tendency to activate prior to hippocampal SWRs is greater than chance (n = 14; one-sided one-sample Wilcoxon signed-rank test; the median is greater than 0.5 with p=$1.831 \times 10^{-4}$). (**iii**) Representative peri-SWR RSC activity sorted by AI calculated for each individual peri-SWR RSC trace. Color bar represents z-scored iGluSnFR signal. The black trace shows the mean peri-SWR RSC iGluSnFR signal. Note that the chance of neocortical activation preceding SWRs is higher than following them. (**B**) (**i**) Distribution of RSC AI values for a representative animal. Dashed lines indicate the first ($Q_1$) and third quartiles ($Q_3$). The SWRs for which the associated RSC AI values are less and greater than $Q_1$ and $Q_3$ are called RSC QR1 and QR4, respectively. RSC QR2-3 consists of all other SWRs. (**ii**) Example plots of mean peri-SWR RSC iGluSnFR signal associated with RSC QR1 (blue) and QR4 (red). The thickness of the shading around each plot indicates SEM. Note that the activity associated with RSC QR1 and QR4 peak before and after SWR centers, respectively. (**iii**) Time course of exemplar mean peri-SWR hippocampal MUA associated with SWRs in RSC QR1 (blue) and RSC QR4 (red). Notice that both hippocampal MUA activity and RSC iGluSnFR activity are negatively (negative AI) and positively (positive AI) skewed for RSC QR1 and QR4, respectively. (**iv**) Distributions of hippocampal MUA AI values for SWRs in RSC QR1 (blue) and QR4 (red) in a representative animal. The vertical line represents the median of each distribution. (**v**) Summary of median values calculated in (iv) across all animals (n = 14, one-sided paired Wilcoxon signed-rank test; RSC QR1 versus RSC QR4 p=$6.103 \times 10^{-5}$). This figure has three figure supplement.

The online version of this article includes the following source data and figure supplement(s) for figure 5:

**Source data 1.** Comparing HPC MUA AI median values across RSC QR1 and RSC QR4.
**Figure supplement 1.** Asymmetry Index is a more robust measure of activation latency compared to peak activation timestamp.
**Figure supplement 2.** Neocortical activation latency relative to SWRs is similar under sleep/urethane anesthesia and VSD/iGluSnFR imaging conditions.
**Figure supplement 3.** A delayed neocortical activation after SWRs in RSC QR1 was not observed.

or above the third quartile (RSC QR4) (*Figure 5B*). Interestingly, when neocortical activity tended to precede the SWRs there was a buildup of hippocampal MUA preceding the SWRs as well, whereas when the neocortex activation followed the SWRs, the hippocampal MUA likewise followed. In other words, the HPC AI values for SWRs in RSC QR1 were less than those for SWRs in RSC QR4, and the HPC AI values for SWRs in RSC QR1 and RSC QR4 were more negative and positive, respectively (*Figure 5Bii–v*; n = 14 per group in *Figure 5Bv*; one-sided paired Wilcoxon signed-rank test; RSC QR1 versus RSC QR4 p=6.103 × 10$^{-5}$). This pattern suggests that the hippocampus and neocortex may engage in waves of mutual excitation in which a SWR may occur at any time before, during or after the peak of the hippocampal-neocortical activation.

## Skewness of peri-SWR hippocampal MUA informs the neocortical activation latency relative to SWRs

We then investigated the inverse question, whether skewness of hippocampal MUA is a SWR-associated feature that correlates with whether neocortical activity precedes or follows SWRs. To do so, AI was calculated for peri-SWR hippocampal MUA traces, and the corresponding calculated AIs were referred to as HPC MUA AI (*Figure 6Ai*). Then, the proportion of SWRs whose RSC and HPC MUA AI values matched in sign (indicating either both preceded or both followed SWRs) were calculated.

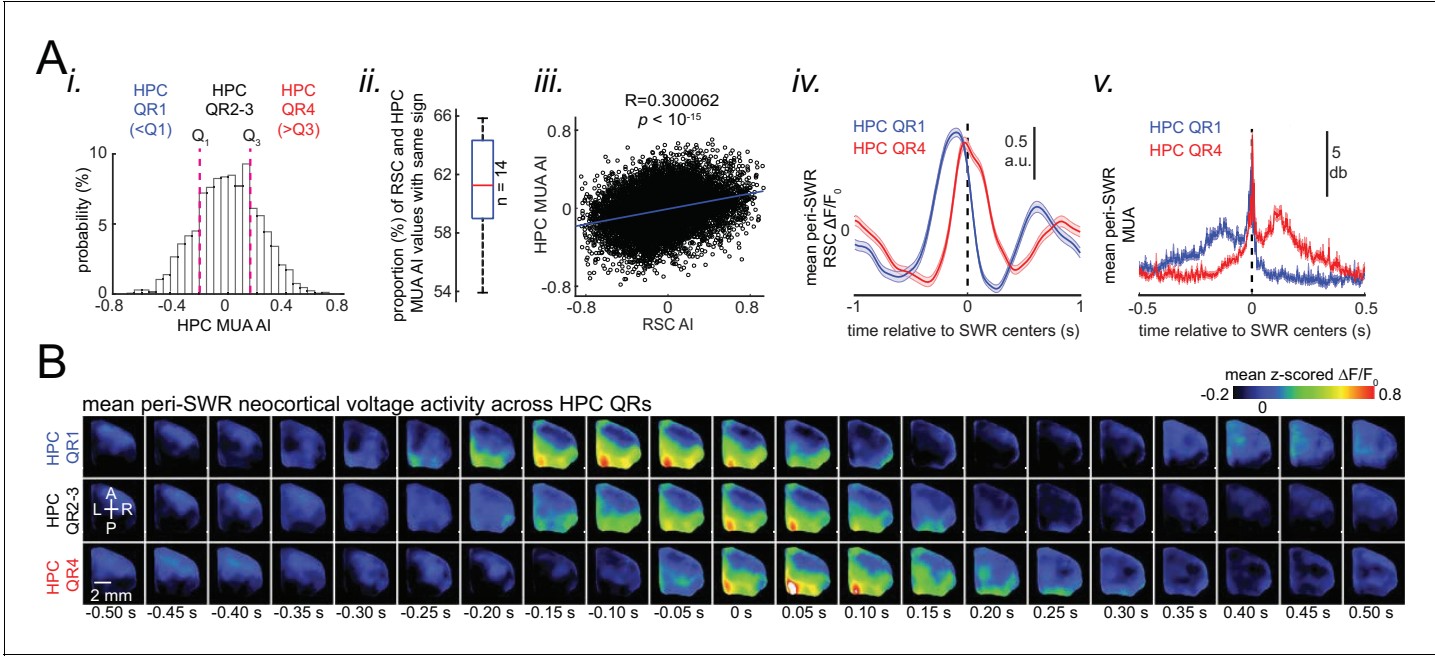

**Figure 6.** Skewness of peri-SWR hippocampal MUA informs neocortical activation latency relative to SWRs. (**A**) (i) Distribution of hippocampal MUA AI values for a representative animal. Dashed lines (Q₁ and Q₃) indicate the first and third quartiles. The SWRs for which the associated hippocampal MUA AI values are less than Q₁ and greater than Q₃ are called HPC QR1 and QR4, respectively. HPC QR2-3 consists of all other SWRs. (ii) Distribution of the proportion of SWRs in each animal for which the sign of both RSC and hippocampal MUA AI values match, as an indication of how well hippocampal MUA can inform whether RSC activity precedes or follows SWRs. The horizontal red line indicates the median of all proportion values in n = 14 animals (significantly above the chance level of 50%; two-sided Wilcoxon signed rank test p=1.220703125 × 10–4). (iii) Correlation between the RSC and hippocampal MUA AI values pooled across all animals (14 animals) is low but significant (n = 11725 SWRs across all animals; two-sided t-test p<10$^{-15}$). (iv) Example plots of mean peri-SWR RSC glutamate activity associated with HPC QR1 (blue) and QR4 (red). (v) Time course of exemplar mean peri-SWR HPC MUA traces associated with SWRs in HPC QR1 (blue) and Q4 (red). Notice that both HPC MUA activity (v) and RSC glutamate activity (iv) are negatively skewed (negative AI) for HPC QR1, and the converse is true for HPC QR4. (**B**) Mean neocortical voltage activity centered on SWR centers associated with HPC QR1, QR2-3, and QR4 in a representative animal. Note the relative temporal shift in activity across three quartile ranges. This figure has two figure supplement.

The online version of this article includes the following figure supplement(s) for figure 6:

**Figure supplement 1.** Whether retrosplenial cortex activation precedes or follows SWRs is correlated by skewness peri-SWR hippocampal MUA.

**Figure supplement 2.** The correspondence between neocortical activation latency relative to SWRs and skewness of peri-SWR hippocampal MUA is similar under sleep/urethane anesthesia and VSD/iGluSnFR imaging conditions.

We found that, on average, only ~ 61% of SWRs have sign-matched RSC and HPC MUA AI values (*Figure 6Aii*; significantly above the chance level of 50%; two-sided Wilcoxon signed rank test p=1.220703125 × 10$^{-4}$). Moreover, the correlation coefficient between HPC MUA and RSC AI values was low but significant (r = ~0.3; n = 11725; two-sided t-test p<10$^{-15}$; *Figure 6Aiii*). We then partitioned the distribution of HPC MUA AI values into lower and higher quartile ranges according to whether the HPC MUA AI range fell below the first quartile (HPC QR1) or above the third quartile (HPC QR4) (*Figure 6Ai*). We found that SWRs, preceded or followed by MUA (HPC QR1 vs QR4), were similarly preceded or followed by RSC activity, respectively (*Figure 6Aiv-v*; *Figure 6—figure supplement 1*). The effects observed for RSC activity can be generalized to other neocortical regions (*Figure 6B*). Thus, these results suggest that elevation of hippocampal MUA is correlated with neocortical activation independent of the timestamp at which SRWs occur.

## Occurrence of single/isolated ripples versus ripple bundles correlates with whether RSC activation precedes or follows hippocampus

Although the skewness of peri-SWR MUA informs the variations in neocortical activation latency relative to SWRs, it does so to a weak extent (*Figures 5B* and *6A*), suggesting that additional factors, other than skewness of peri-SWR HPC MUA, are required to predict whether neocortex precedes or follows hippocampus around SWRs. We found that the occasional occurrence of 'bundle' of two or more SWRs (*Davidson et al., 2009*; *Wu and Foster, 2014*) was another factor correlating with peri-SWR neocortical activity latency (*Figure 7*; *Figure 7—figure supplement 1*; *Figure 7—video 1*). On average, neocortex stayed active for a longer time and peaked later around the bundled than single/isolated ripples (*Figure 7—figure supplement 1A–C*). This observation suggests that there is a higher chance for the neocortex to follow the bundled ripples compared to the single/isolated ones (*Figure 7B–E*; *Figure 7—figure supplement 1C*). Interestingly, the neocortical deactivation preceding SWRs was stronger in bundled than isolated ripples, pointing to the strength of neocortical down-states as a predictor of occurrence of bundled ripples (*Figure 7—figure supplement 1Dii*). Moreover, the neocortical activation was stronger around bundled than isolated ripples (*Figure 7—figure supplement 1Cii*).

Although short inter-SWR intervals leading to 'bundles' of ripples can simply be a result of an underlying Poisson process, it is possible that bundled ripples constitute a random process different from that of single/isolated events. If the former is true, then the inter-SWR interval should be an exponential distribution (*Shen and McNaughton, 1996*). The inter-SWR time interval distribution of SWRs occurrence was well modeled by an exponential curve (linear curve in logarithmic scale) only for intervals longer than ~ 250 ms (single/isolated ripples). However, for shorter intervals (bundled ripples), there was an excess of positive residuals of the fitted exponential curve (*Figure 2—figure supplement 2E*). This suggests that the mechanisms of SWR generation when they occur in close temporal proximity to each other obeys a non-Poisson dynamic, constituting a different random process from that of single/isolated ripple events.

## Discussion

The interactions between neocortex and hippocampus play an integral role in memory consolidation during sleep. These interactions are particularly clear during SWRs, where replays of both hippocampal and neocortical memory-associated traces tend to co-occur (*Jadhav et al., 2016*; *Ji and Wilson, 2007*; *Peyrache et al., 2009*). Therefore, it is important to have a functional map of peri-SWR neocortical activity during sleep, which could inform our search for neocortical regions engaged in memory consolidation. In this study, we partly filled this gap by providing a mesoscale spatiotemporal map of peri-SWR dorsal neocortical activity.

### Patterns of activity in neocortical regions are differentially modulated around hippocampal SWRs

A differential level of activation and deactivation around SWRs was observed across neocortical regions. This differential modulation was correlated with neocortical structural connectivity: regions with strong axonal interconnections were co-modulated to a similar extent around SWRs. Although significant peri-SWR modulation of discrete neocortical regions has been reported previously using electrophysiological methods, for example, in prefrontal (*Battaglia et al., 2004*; *Mölle et al., 2006*;

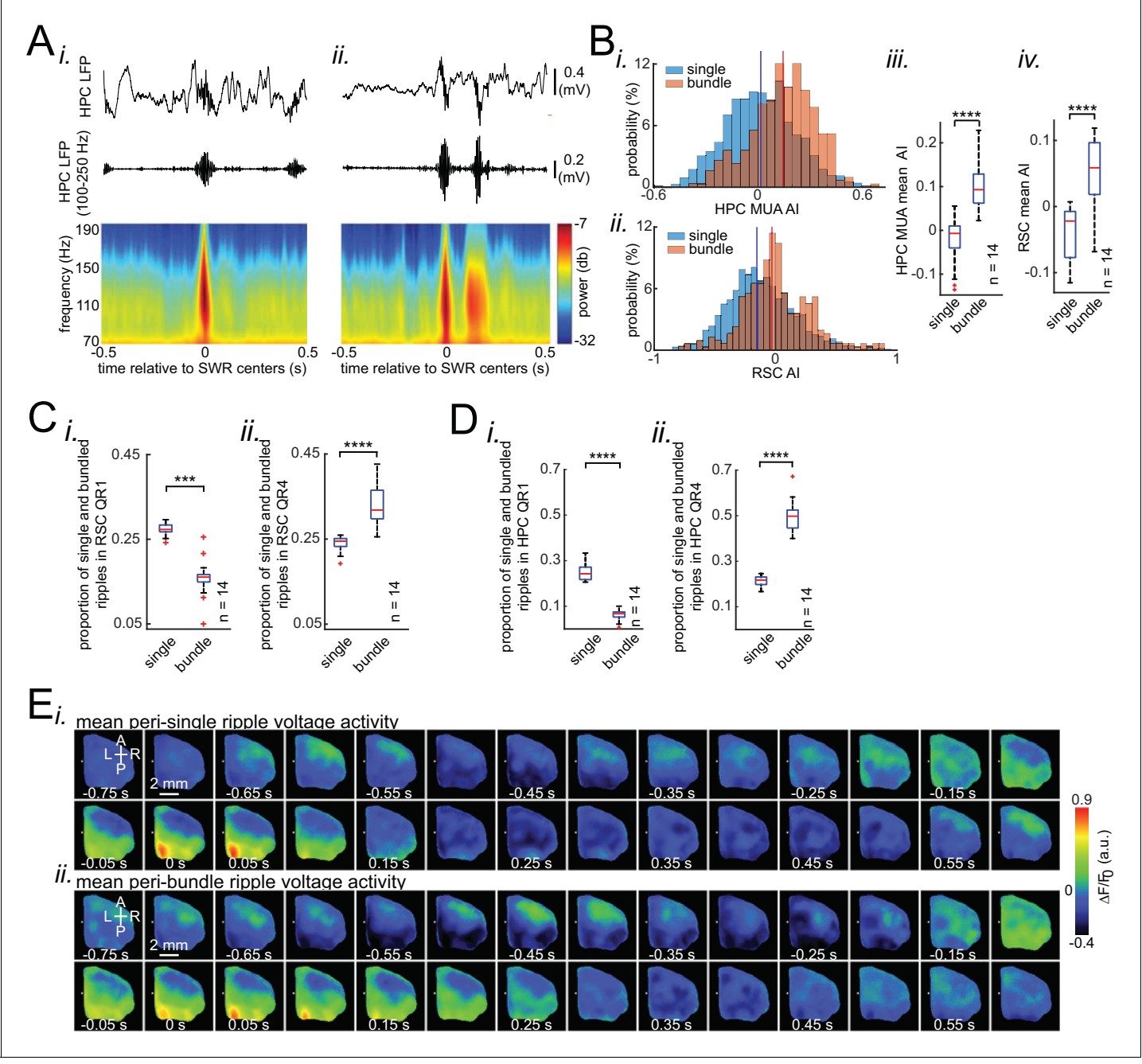

**Figure 7.** Occurrence of single/isolated ripples versus ripple bundles correlates with whether RSC activation precedes or follows hippocampus. (**A**) Example raw hippocampal LFP signal (top row), ripple-band filtered signal (middle row), and mean peri-SWR spectrogram (bottom row) displaying (i) single/isolated and (ii) bundled ripples. (**B**) Distribution of HPC MUA (i) and RSC (ii) AI values calculated for single/isolated (blue) and bundled (red) ripples in a representative animal. The blue and red vertical lines represent the means of blue and red distributions, respectively. As expected, the red distribution is shifted to the right with respect to the blue one. (iii-iv) Comparison of HPC MUA (iii) and RSC (iv) AI mean values for single/isolated and bundled ripples (n = 14; one-sided paired Wilcoxon signed-rank test; in iii p=6.103 × $10^{-5}$; in iv p=6.103 × $10^{-5}$). The red plus signs represent the data points outside the interval (centered on median value) which includes 99.3 percent of all data points. Red horizontal lines represent the medians. (**C–D**) Summary data from n = 14 mice representing the proportion of single/isolated versus bundled ripples fell in RSC (**C**) and HPC (**D**) QR1 (i) and QR4 (ii). Note that there is a higher chance for single/isolated ripples to lie in RSC or HPC QR1 and higher chance for bundled ripples to fall in RSC or HPC QR4 (one-sided paired Wilcoxon signed-rank test; in C p=1.221 × $10^{-4}$; in D p=6.103 × $10^{-5}$). (**E**) (i–ii) Representative montages of mean neocortical voltage activity centered on single/isolated ripples (i) and the first ripple in bundled ripples (ii). Notice that neocortex stays active longer and peaks later around bundled versus single/isolated ripples. Moreover, there is strong neocortical deactivation preceding bundled ripples. This figure has two figure supplement.

*Figure 7 continued on next page*

*Figure 7 continued*

The online version of this article includes the following video, source data, and figure supplement(s) for figure 7:

**Source data 1.** Comparing HPC MUA AI mean values across single and bundled ripples.
**Source data 2.** Comparing RSC AI mean values across single and bundled ripples.
**Source data 3.** Comparing the proportion of single versus bundled ripples in RSC QR1.
**Source data 4.** Comparing the proportion of single versus bundled ripples in RSC QR4.
**Source data 5.** Comparing the proportion of single versus bundled ripples in HPC QR1.
**Source data 6.** Comparing the proportion of single versus bundled ripples in HPC QR4.
**Figure supplement 1.** Distinct latency, strength, and duration of neocortical modulation around single/isolated versus bundled ripples.
**Figure supplement 2.** The correlation between occurrence of single/isolated versus bundled ripples and latency of peri-SWR RSC activation is similar under sleep/urethane anesthesia and VSD/iGluSnFR imaging conditions.
**Figure 7—video 1.** Mean peri-single-SWR versus peri-bundled-SWR voltage activity under urethane anesthesia.
https://elifesciences.org/articles/51972#fig7video1

*Peyrache et al., 2011*; *Peyrache et al., 2009*), posterior parietal (*Wilber et al., 2017*), entorhinal (*Isomura et al., 2006*), primary somatosensory (*Sirota et al., 2003*), visual (*Ji and Wilson, 2007*), and auditory corticex (*Rothschild et al., 2017*), previous studies were unable to simultaneously characterize the differential patterns of activity across such regions, due to a lack of extensive spatial coverage of the neocortical mantle. *Battaglia et al. (2004)* however, have mentioned a midline bias in the modulation of firing rate of different neocortical regions around SWRs, but they left the detailed analysis of such regional effect for future studies. Here, we expanded upon their finding by utilizing high spatiotemporal resolution wide-field optical imaging of voltage and glutamate activity combined with electrophysiology.

The only study to our knowledge that has compared the modulation of neocortical regions around SWRs (*Logothetis et al., 2012*) reported a significant up-regulation of Blood Oxygen Level-Dependent (BOLD) signal in almost the entire neocortex around SWRs in monkeys, whereas we observed a dichotomy among neocortical regions where somatosensory regions did not show a significant modulation compared to medial subnetwork regions. The previous study also reported a down-regulation of mean peri-SWR BOLD signal in primary visual cortex (*Logothetis et al., 2012*), which is in contrast to our results in which V1, along with other visual regions, showed strong activations. These discrepancies could be due to an intrinsic difference between BOLD and optical voltage-sensitive dye (VSD) and iGluSnFR imaging signals. As opposed to optical VSD and iGluSnFR signals, BOLD signal is an indirect, and substantially slower measure of neuronal electrical activity, which sometimes may not faithfully reflect the underlying neuronal activity (*Iadecola and Nedergaard, 2007*). On the other hand, mesoscale optical imaging detects activity predominantly in the superficial ~ 300 microns of neocortex, although this layer does contain dendrites from lower layers. Additionally, this discrepancy could be due to the difference in animal model (macaque vs mouse), although this seems relatively less likely given the increasingly documented connectivity similarities between these species.

The subnetworks that were accessible to us for imaging are subsets of larger brain-wide networks. In particular, a medial subnetwork, consisting of retrosplenial and posterior parietal cortices, is a subset of the default mode network which is involved in several cognitive processes including memory (*Smith et al., 2018*; *Stafford et al., 2014*). Therefore, our results suggest that default mode network is probably one of the most modulated networks of the mouse brain around SWRs. This idea has been supported in macaques (*Kaplan et al., 2016*).

The reported gradation in modulation extent of different neocortical regions begs the question of what is special about visual sensory areas, which were the second most modulated regions around SWRs, compared with somatosensory areas, which were the least modulated regions. The answer to this question could be related to the stronger structural connectivity between retrosplenial and visual areas compared with retrosplenial and somatosensory cortices (*Sugar et al., 2011*; *Van Groen and Wyss, 2003*). However, it does not completely settle the problem. Assuming that neocortical activation around SWRs has a mnemonic function, one may plausibly expect a stronger activation in neocortical regions that are more involved in daily life information processing. For example, vibrissal sensation is highly active during wakefulness and probably plays an important role in memory processes in the rodent brain. But, why is it the case that barrel cortex is not as active as visual cortices

around SWRs? It is an interesting question that can be addressed in a separate study with using appropriate behavioral tasks. For instance, the peri-SWR differential activation of neocortical regions after performing odor-memory and visually-guided spatial memory tasks could be compared.

## Ripple power is distinctively correlated with peak activity in different neocortical subnetworks

There was a differential association between peak activity in different neocortical regions/subnetworks and hippocampal ripple power. A similar pattern as the one observed earlier in peri-SWR averaging analysis also appeared here, where there was a significantly stronger association between hippocampal ripple power and peak activity in medial compared to somatomotor subnetworks. *Mölle et al. (2006)* reported a relevant finding, that the probability of ripple occurrence was locked to mPFC EEG positive peak (depth LFP negative peak; equivalent to activation in this paper). They observed the lowest followed by the highest SWR occurrence probabilities well before and very close to the EEG positive peak, respectively.

We also know from other works that there is a significant association between neocortical up-/down-states and SWR occurrence (*Battaglia et al., 2004*; *Isomura et al., 2006*; *Mölle et al., 2006*; *Peyrache et al., 2011*; *Peyrache et al., 2009*; *Sirota et al., 2003*), although many up-states are not associated with SWRs. Consistent with previous reports, we also found a strong modulation of hippocampal ripple power around the onset of neocortical up-/down-states. Moreover, we extended the previous findings by providing the spatial distribution of correspondence between neocortical up-states and hippocampal ripple power.

## Neocortex tends to activate sequentially from medial to more lateral regions around SWRs

There was a medial to lateral temporal gradient of activation of neocortical regions around SWRs on a time scale of about 30 ms. To the best of our knowledge, such mediolateral sequential activation of neocortical regions around SWRs has not been previously reported, probably because older studies used electrophysiological techniques with sparser spatial coverage. However, *Logothetis et al. (2012)*, in their pioneering study in monkeys, reported a sequential activation among neocortical regions on a time scale of a few seconds using fMRI, where temporal, frontal and prefrontal regions activated earlier than sensory areas around SWRs. In our imaging window, we did not have access to the prefrontal and temporal areas especially medial prefrontal cortex, medial and lateral entorhinal cortices whose activities modulation around SWRs has already been studied (*Isomura et al., 2006*). Nonetheless, we did observe that visual and auditory sensory cortices were the latest among neocortical regions to activate.

Two major opposite directions of propagation for neocortical waves of activity during slow-wave sleep (SWS) in mice have been reported, posterior-medial to anterior-lateral and anterior-lateral to posterior-medial directions (*Greenberg et al., 2018*; *Mohajerani et al., 2013*; *Mohajerani et al., 2010*). Moreover, retrosplenial cortex has been shown to have the highest probability to be the initial zone of waves with the former direction (*Greenberg et al., 2018*). Therefore, our results suggest that SWRs tend to occur during slow waves with the posterior-medial to anterior-lateral direction of propagation. It is probable that the neocortical up-states occurring in the absence of SWRs tend to have the opposite direction of propagation as reported in human studies (*Massimini et al., 2004*), but we did not investigate this idea in the present work.

In addition, compatible with previous reports, our data show that almost all neocortical regions tend to start activating more or less before the hippocampus generates a SWR (*Figure 4Aii*), which suggests a neocortex-to-hippocampus initial direction of information flow around SWRs. It could be interpreted from the viewpoint of hippocampal memory indexing theory in this way: an initial flow of incompletely retrieved memories from neocortex to hippocampus could lead to reactivation of a corresponding memory index code, stored in hippocampus, which in turn may lead to reactivation of global neocortical memory traces. The potential biasing of hippocampal reactivation by neocortical one has been previously proposed (*Helfrich et al., 2019*; *Klinzing et al., 2019*; *Ngo et al., 2019*; *Norman et al., 2019*; *Rothschild, 2019*; *Rothschild et al., 2017*; *Sirota et al., 2003*; *Wang and Ikemoto, 2016*). Therefore, RSC, which is the first among dorsal neocortical regions to activate around SWRs, may mediate this initial information flow, probably through entorhinal cortex, and

subsequently bridge the broadcast of hippocampal index code to the rest of the neocortical mantle. In fact, the pattern of structural connectivity of RSC with entorhinal cortex and hippocampus (*Wyss and Van Groen, 1992*) supports this idea, where numerous efferents from RSC to entorhinal cortex, the principal input zone to hippocampus, and ample afferents from subicular structure, the main output gate in hippocampus, to RSC have been reported (*Amaral and Witter, 1989*).

The neocortical-hippocampal-neocortical loop of information flow has been observed around hippocampal SWRs after an extensive training in spatial tasks (*Rothschild, 2019*; *Rothschild et al., 2017*). One way of detecting such a potential loop in our data was to check if the neocortical activations that precede SWRs (RSC QR1 in *Figure 5Bii*) also lead to delayed neocortical activations. We tested this idea and did not observe a delayed neocortical activation (*Figure 5—figure supplement 3*). However, it does not mean the peri-SWR neocortical-hippocampal-neocortical loop of information flow does not exist. The lack of neocortical-hippocampal-neocortical loop in our data could be explained by the extent of animal experience prior to data collection. *Rothschild et al. (2017)* trained rats for at least 12 days in a spatial task, while the mice used in our experiment were naïve. Although unlikely, the lower temporal resolution of glutamate and voltage imaging, compared with that of single-unit activity presented in Rothschild et al. may also explain why we did not observe a neocortical-hippocampal-neocortical loop in our data.

## Skewness of peri-SWR hippocampal MUA informs neocortical activation latency relative to SWRs

Two general patterns of hippocampal MUA around SWRs were observed, a gradual build-up leading to a sharp transient increase at the SWR time and a sharp transient increase at the SWR time followed by a gradual decrease to baseline (*Figure 5Bii–v*). Interestingly, these two leading and lagging patterns of hippocampal MUA corresponded to two leading and lagging patterns of activity in the neocortex, respectively.

*Ji and Wilson (2007)* reported periods of sustained MUA interleaved by periods of silence in both neocortex and hippocampus during SWS, which could be a reflection of slow oscillations. They called these periods of sustained activity 'frames'. They found that neocortical and hippocampal frames tend to co-occur and neocortical frames, on average, tend to precede hippocampal ones by ~ 50 ms. They also reported that there is a higher ripple power right after the onset and right before the offset of the hippocampal frames compared to other times. Moreover, ripple power was higher at the end compared to the beginning of hippocampal frames. Taking into account the co-occurrence of neocortical and hippocampal frames, we can infer that ripple power is probably higher at the end compared to the beginning of neocortical frames as well, potentially denoting a higher probability of ripple occurrence at the end of neocortical up-states. This inference is compatible with our results, reported in *Figure 3—figure supplement 2*, where we observed an elevation of hippocampal ripple power right before the onset of RSC down-states.

We speculate that the observed temporal diversity in hippocampal-neocortical interactions around SWRs could reflect the degree of consolidation of a given memory, where neocortex leads hippocampus for well-consolidated memories and lags for newly formed ones. Our prediction is that reactivation of recently formed memory traces tend to happen during SWRs around which neocortex follows hippocampus. Since the timing of neocortical peak activity can be inferred from hippocampal MUA skewness, it is not necessary to directly record neocortical activity to test this hypothesis. Datasets with only hippocampal recordings could be assessed for this purpose.

We should emphasize that the propagating nature of SWRs (*Patel et al., 2013*) does not influence the results reported in the current study. The majority of SWRs propagate less than 3 mm across the septal-intermediate or intermediate-temporal portion of the rat hippocampus with a minimum speed of 0.33 mm/ms (*Patel et al., 2013*). Considering the lengths of mouse hippocampus septal-intermediate and intermediate-temporal portions (~3.25 mm) (*Milior et al., 2016*), it would take less than 9 ms (equivalent to one cycle of ripples) for SWRs to reach to the electrode tip from their initiation zones. Therefore, changing the location of the electrode tip in the septal-intermediate portion of the hippocampus would have a minimal effect on the results reported in this study. However, our findings could have been different had we recorded from the intermediate-temporal portion of HPC since SWRs in dorsal and ventral portions of the hippocampus often do not co-occur (*Patel et al., 2013*).

## Occurrence of single/isolated versus bundled ripples correlates with whether RSC activation precedes or follows hippocampus

Consistent with previous findings (*Davidson et al., 2009*; *Wu and Foster, 2014*), we observed that multiple SWRs could sometimes occur in close proximity as a bundle of ripples. We found that, on average, the neocortical tendency to follow hippocampus is higher around bundled compared to isolated ripples. We also observed that all neocortical subnetworks except the medial one were activated more strongly around bundled compared to isolated ripples.

It has been reported that reactivation of prolonged sequences of place cells, encoding recently traversed long routes by rats, tends to occur during bundled ripples (*Davidson et al., 2009*; *Wu and Foster, 2014*). An isolated ripple has a limited duration and therefore its neural content consists of a sequence of limited number of place cells with place fields covering routes with a limited length (*Davidson et al., 2009*). Therefore, if coordinated interaction between hippocampus and neocortex is essential for memory consolidation, it is expected that neocortex stays active longer during bundled ripples, when the hippocampus is reactivating memories of larger content and/or longer duration. This is what was observed, suggesting bundled ripples tend to occur during longer neocortical up-states when there is a longer temporal window available for a potential hippocampal-neocortical communication. Interestingly, according to our data, the probability of occurrence of a bundled ripple in hippocampus is correlated with the pre-SWR (not peri-SWR) deactivation amplitude in neocortex. This dip could be a signaling mechanism for an upcoming longer communication between the two structures.

In conclusion, this work sheds light on the dynamics of hippocampal-neocortical interactions around SWRs which are thought to underlie system memory consolidation processes. Our results reveal neocortical hotspots mediating the broadcast of hippocampal representations to the rest of neocortex by identifying retrosplenial cortex as a potential bridge between these two structures. Past the RSC, information flow tends to be carried along a medial-lateral direction, demonstrating the order of recruitment of different neocortical modules, potentially associated with different features of given memory, around SWRs. Lastly, neocortical activation latency around SWRs is informed by hippocampal features including MUA latency as well as the number of ripple events. The observed spectrum of temporal latencies in activation of hippocampus and neocortex around SWRs possibly reflects the direction of communication during hippocampal and neocortical reactivations of recent versus remote memories. We predict that encoding of new hippocampal-dependent memories lead to a transient bias toward hippocampus leading neocortex around post-encoding sleep SWRs and that this bias would last until the load of consolidating new information gradually dissipates.

## Materials and methods

### Animals

A total of twelve mice, female and male, were used for the imaging studies. For natural sleep experiments, four adult (>2 months) iGluSnFR transgenic mice (strain Emx-CaMKII-Ai85), expressing iGluSnFR in glutamatergic neocortical neurons (*Marvin et al., 2013*; *Xie et al., 2016*), were used. We generated Emx-CaMKII-Ai85 transgenic mice by crossing the homozygous B6.129S2-Emx1tm1 (cre)Krj/J strain (Jax no. 005628) and the B6.Cg-Tg(CamK2a-tTA)1Mmay/DboJ strain (Jax no.007004) with the hemizygous B6;129S-Igs7 tm85(teto-gltI/GDP*)Hze/J strain (Jax no.026260). This crossing is expected to produce expression of iGluSnFR within all excitatory neurons across all layers of the cortex, but not in GABAergic neurons (*Huang and Zeng, 2013*; *Madisen et al., 2015*). Brain sections of the positive transgenic mice confirmed robust expression in the neocortex and hippocampus (*Figure 1—figure supplement 1*). At the end of natural sleep experiments, two of the iGluSnFR mice were anesthetized with urethane and imaged. Eight C57BL/6J mice from Jackson Laboratory were also used for acute voltage-sensitive dye imaging under urethane anesthesia. An additional set of 6 C57BL/6J mice, including both males and females were used for the electrophysiological studies. Mice were housed in groups of two to five under a 12 hr light-dark cycle. Mice were given ad libitum access to water and standard laboratory mouse diet at all times. Mice used in the sleep recording experiments were housed singly after head-plate/electrode implantation surgery. The animal

protocols were approved by the University of Lethbridge Animal Care Committee and were in accordance with guidelines set forth by the Canadian Council for Animal Care.

## Surgery

### Urethane surgeries

Animals were administered with urethane (1.25 mg/kg) and immediately anesthetized with isoflurane (1–2% mixed in $O_2$) in order to perform the surgery. We began to administer urethane before the isoflurane, so that by the time we reached the recording stage, urethane would be in effect. This order of administration improved surgery success and reduced fatality rate. We were gradually reducing isoflurane levels over the course of surgery, and isoflurane levels diminished completely by the time urethane recordings began. After animals were anesthetized with isoflurane, mice were placed on a metal plate that could be mounted onto the stage of the upright macroscope, and the skull was fastened to a steel plate. Before craniotomy, the electrode was implanted in the pyramidal layer of dorsal hippocampus. A 7 × 6 mm unilateral craniotomy (bregma 2.5 to −4.5 mm, lateral 0 to 6 mm) was made, and the underlying dura was removed, as described previously. Body temperature was maintained at 37°C using a heating pad with a feedback thermistor during the surgery and recording sessions. Animals were also given a tracheotomy to reduce breathing complications.

### Head-fixed natural sleep surgeries

For natural sleep experiments, we performed subcutaneous injections with 0.5 gr/Kg buprenorphine half an hour before the surgery started. Animals were then anesthetized with isoflurane (1–2% mixed in $O_2$). Following anesthesia, we performed the following: removed the head skin, implanted the hippocampal LFP electrode, implanted a head-plate, covered the skull with a thin and transparent layer of the metabond (Parkell, Inc), and at the end covered the skull with a glass coverslip. An additional bipolar electrode was also implanted in the neck muscles for recording EMG activity. Animals were allowed to recover for two weeks before recordings started.

### Surgeries for unrestrained natural sleep experiments

The animals were anesthetized with isoflurane (2.5% induction, 1–1.5% maintenance), and after removing the skin, multiple holes with particular coordinates were drilled on the skull for implanting hippocampal and neocortical electrodes. An additional bipolar electrode was also implanted in the neck muscles for recording EMG activity. For neocortical and hippocampal recordings of LFPs, bipolar (tip separation = 0.6 mm) and monopolar electrodes, made from Teflon-coated stainless-steel wire with bare diameter of 50.8 µm (A-M Systems), were implanted in several neocortical areas as well as the pyramidal layer of CA1 in dorsal hippocampus according to the following coordinates in mm: primary motor cortex (M1): AP: 1.5, ML: −1.7, DV: 1.5, secondary motor cortex (M2): AP: 1.7, ML: 0.6, DV:1.1 mm, mouth primary somatosensory area (S1M): AP: 0.85, ML: 2.8, DV: 1.4, barrel primary somatosensory area (S1BC): AP: −0.1, ML: −3.0, DV: 1.4 mm, retrosplenial cortex (RSC): AP: −2.5, ML: 0.6, DV: 1.1 mm, and CA1 subfield of hippocampus (HPC): AP: −2.5, ML: 2.0, DV:1.1 mm. For EMG, a multi-stranded Teflon-coated stainless-steel wire with bare diameter of 127 µm (Cooner Wire) was implanted into the neck musculature using a 22 gauge needle. The reference and ground screws were placed on the skull over the cerebellum. The other end of all electrode wires were clamped between two receptacle connectors (Mill-Max Mfg. Corp.). Clamped connectors were fixated on the skull using metabond (Parkell, Inc) and dental cement. Body temperature was maintained at 37°C using a heating pad with a feedback thermistor during the surgery. Animals were allowed to recover for one week before recordings started.

## Habituation for head-restraint natural sleep experiments

After 14 d of recovery, mice were habituated to the recording setup, by putting the animals one by one on the recording platform with one or two pieces of Cheerios cereal. They were allowed to explore the platform and acclimatize to the environment. They gradually started to eat the cheerios after a few days. Then the animals were head-restrained with daily prolongation of head-fixation duration. We started with five minutes on the first day and increased the head-restrained duration by five minutes per day for up to one hour. During head-fixation period, each animal was loosely surrounded by a plastic tube to limit motion and guide relaxation. A day before recording, the animals

were moved from their home cage colony room to another room at noontime. We prevented them from falling asleep for around nine hours while they were kept in their own home cage by touching them using a cotton-tip stick whenever they showed signs of falling asleep. In the final hours of sleep restriction, we sometimes gently handled the animals. Then, we transferred the mice to a bigger cage containing new objects, including a running wheel and many pieces of cheerios and a water container. We left the mice in the new cage in the same room overnight. The next day, the mice were transferred to the recording room early in the morning and the recording initiated at around 6:00 AM. After the recording, mice were allowed to recover and sleep at will, in their own home cage in the colony room, for at least three days before repeating this procedure for subsequent recording sessions.

The main reason for not conducting all the experiments under natural sleep was the low success rate of having animals sleeping under head fixation, especially with a slightly tilted head position. Tilting the head was done to include temporal cortices (e.g. the auditory cortex) within our imaging window. Therefore, we used a model of sleep, that is urethane anesthesia, to make our sample size larger.

## Neocortical and hippocampal LFP recording in head-restrained imaging experiments

Teflon coated 50 μm stainless steel wires (A-M Systems) were used for the hippocampal and neocortical LFP and MUA recordings. For HPC recordings, we drilled a hole on the right hemisphere skull about 2.6 mm lateral to the midline and tangent to the posterior side of the occipital suture. Then, the tip of the monopolar electrode was gradually lowered through the hole at a 57 degree angle with respect to the vertical axis perpendicular to the surface on which the stereotaxic apparatus was sitting. During lowering, the electrode signal with respect to the reference electrode located on top of the cerebellum, was monitored both visually and audibly. Lowering the electrode was stopped as soon as a dramatic increase in the MUA was heard and observed near the calculated coordinate (angle = 57 degrees, depth = ~1.75 mm) for the pyramidal layer of dorsal CA1. After identifying the optimum location for the electrode tip, we fixated the electrode on the skull using Krazy Glue and dental cement. For the neocortical LFP and MUA recordings, the tip of the electrodes were placed in the superficial layers of neocortical regions (usually motor cortex) at the edge of the cranial window. The electrode signals were amplified (x 1,000) and filtered (0.1–10000 Hz) using a Grass A.C. pre-amplifier Model P511 (Artisan Technology Group , IL) and digitized using a Digidata 1440 (Molecular Device Inc, CA) data acquisition system at 20 kHz sampling rate. For the chronic natural sleep experiments, we used the same wire but in the bipolar form to record hippocampal LFP. After data collection, 1 mA positive and negative currents were injected into the hippocampal electrode for about 200 ms to mark electrode location for histology. Animals were perfused with PBS (1x) and PFA (4%) and their brains were extracted, sectioned and mounted. Location of the hippocampal electrode was further confirmed postmortem using cresyl violet staining (*Figure 1—figure supplement 1*).

## Neocortical and hippocampal LFP recording in unrestrained sleep experiments

Animals were left undisturbed to recover for 7 days after surgery and then habituated for 5–7 days in the recording chamber. On recording days, animals were transferred to the recording chamber where their sleep activity was recorded starting from 8:30 AM for 4 hr using a motorized commutator (NeuroTek Innovative Technology Inc, On, Canada). LFPs and EMG activity were amplified, filtered (0.1–4000 Hz) and digitized at 16 kHz using a Digital Lynx SX Electrophysiology System (Neuralynx Inc, MT) and the data were recorded and stored on a local PC using Cheetah software (Neuralynx, Inc MT).

## Voltage sensitive dye imaging

The procedures we used are described previously *Greenberg et al. (2018)*; *Mohajerani et al. (2013)*, and are briefly explained here. After completion of surgery, the exposed brain were incubated with solution containing voltage dye for 30–45 min. The solution was made by dissolving voltage-sensitive dye (VSD) RH-1691 (Optical Imaging, New York, NY) in 4-(2-hydroxyethyl)−1-

piperazineethanesulfonic acid (HEPES)-buffered saline solution (0.5 mg ml$^{-1}$). To minimize the movement artifacts due to the respiration, we covered the stained brain with 1.5% agarose made in HEPES-buffered saline and sealed the cranial window with a glass coverslip. VSD imaging of spontaneous activity in the absence of sensory stimuli began ~ 30 min after removing the dye solution and washing the brain until the residual dye solution was completely removed. Using a charge-coupled device (CCD) camera (1M60 Pantera, Dalsa, Waterloo, ON) and an EPIX E4DB frame grabber with XCAP 3.7 imaging software (EPIX, Inc, Buffalo Grove, IL), we recorded 12-bit images every 10 ms (100 Hz). We used a red LED (Luxeon K2, 627 nm center) and excitation filters of 630 ± 15 nm to excite the VSD. The reflected fluorescent signal from excited dyes was filtered using a 673 to 703 nm bandpass optical filter (Semrock, New York, NY), and was passed through a microscope composed of front-to-front video lenses (8.6 × 8.6 mm field of view, 67 μm per pixel). To reduce the potential artifacts caused by the presence of large neocortical blood vessels, we focused the lens into the neocortex to a depth of ~ 1 mm. Total duration of the VSD excitation in a typical imaging experiment ranged from one to two hours. We also recorded the voltage optical signal from the neocortex in response to different periphery stimulation as described before (*Mohajerani et al., 2013*). Sensory stimulation was used to determine the coordinates for the primary sensory areas (HLS1, FLS1, BCS1, V1 and A1), secondary somatosensory areas (HLS2, FLS2 and BCS2). From these primary sensory coordinates, the relative locations of additional associational areas were estimated using stereotaxic coordinates (ptA, RS, M2, V2M (medial secondary visual cortex), V2L (lateral secondary visual cortex), and M1 (primary motor cortex).

## Glutamate imaging

iGluSnFR was excited with light from a blue-light-emitting diode (Luxeon K2, 473 nm, Quadica Developments Inc, Lethbridge, Alberta) delivered through a band-pass filter (Chroma Technology Corp, 467–499 nm). Ambient light resulting from iGluSnFR excitation (473 nm) was ~ 1.5 mW over the 8 × 8 mm area used for imaging. iGluSnFR fluorescence emission was filtered using a 520–580 nm band-pass filter (Semrock, New York, NY). We collected 12-bit images at 100 Hz using CCD camera (1M60 Pantera, Dalsa, Waterloo, ON) and an EPIX E4DB frame grabber with XCAP 3.7 imaging software (EPIX, Inc, Buffalo Grove, IL). To reduce the potential artifacts caused by the presence of large neocortical blood vessels, we focused the lens into the neocortex to a depth of ~ 1 mm. Total duration of the indicator excitation in a typical imaging experiment ranged from two to three hours. At the last day of experiments, we anesthetized mice with 1.5 gr/kg urethane and imaged the neocortex. We also recorded the iGluSnFR signal in response to different periphery stimulation under urethane anesthesia as described before (*Mohajerani et al., 2013*). Sensory stimulation were used to functionally map the center of the hind-limb somatosensory, fore-limb somatosensory, auditory, visual, and barrel cortices.

## Behavioral and pupil recording

To monitor animal behavior and measure pupil size, we used an IR USB camera (Point Grey Firefly MV USB, FLIR Systems, Inc) with 20 Hertz frame rate. We also used an infrared LED to illuminate the animals' faces and parts of their body including their head, shoulders, and forelimbs.

## Pupil diameter detection

We used an intensity-based thresholding custom made algorithm implemented in Bonsai (https://bonsai-rx.org//) to quantify pupil diameter. Briefly, the algorithm detects the pupil by segmenting it with the surrounding areas of the eye and models it as an ellipse. The main diagonal of the ellipse was considered as the pupil diameter.

## Sleep scoring

Vigilance stages of each animal were classified as awake, NREM, and REM by thresholding EMG and hippocampal theta-to-delta ratio signals (*Niethard et al., 2018*; *Yang et al., 2014*; *Yüzgeç et al., 2018*). Awake periods were identified by visual inspection of animal behavior and EMG signal. NREM sleep was identified as periods with low EMG and theta-to-delta ratio. For head-restrained experiments, we also included a third criteria for detecting NREM sleep periods: lack of facial (around the chin) movement for at least 50 s. REM sleep periods were detected when EMG signals

were the lowest, which is due to muscular atonia, and high theta-to-delta ratios. For head-fixed experiments, we also measured pupil size to verify our sleep scoring results (*Yüzgeç et al., 2018*).

### SWRs detection

We followed the method used in *Mölle et al. (2006)* for detecting ripples. Briefly, the raw hippocampal LFP was first down-sampled to 2 kHz. Then, it was filtered using a 400-order band-pass FIR filter designed in MATLAB (MathWorks). The filtered signal was rectified and smoothed using a rectangular window with a length of 8 ms, generating the ripple power signal. SWRs were identified when the ripple power signal passed the detection threshold defined by the mean plus a multiple (*Figure 2* and *Figure 2—figure supplement 3*: 2 for VSDI and 3 for iGluSnFR imaging; *Figure 2— figure supplement 4*: 3 for VSDI and 4 for iGluSnFR imaging) of standard deviation of the ripple power signals. A lower threshold (75% of the detection threshold) was used to identify the onset and offset of each SWR. Detected events were further screened by applying a duration threshold. Events with a duration shorter than the mean duration of all detected events were excluded. The ripple center was defined as the timestamp of the largest trough between the onset and offset times. Moreover, events with centers less than 50 ms apart were concatenated.

### Splitting SWRs into HPC quartiles

We defined a quantity, called asymmetry index (AI), to measure the asymmetry or skewness of hippocampal MUA and neocortical voltage and iGluSnFR signals centered on SWR centers timestamps (0 s). AI is defined as $(A - B) / (A + B)$, where A and B are the mean values from $-0.2$–0 s and 0–0.2 s respectively. AI values range from $-1$ and 1, where an AI $= -1$ and AI $= 1$ denote complete skewness of the signal to left and right relative to the SWR centers, respectively. The distribution of MUA AI values was split into quartiles. The first and fourth quartiles were utilized for further analyses to examine whether SWRs either preceded or followed by MUA, respectively.

### Bundled ripples detection

Bundled ripples were identified by applying an additional threshold on previously detected SWRs. For each previously detected SWR, the wavelet-based power (using analytic Morlet wavelet) in the ripple-band frequencies (150–250 Hz) was calculated for peri-SWR hippocampal LFP traces. Ripple events were identified as bundled ripples if the following criteria were met: (1) The power signal surpassed an adaptively determined power threshold for at least two successive times; (2) The minimum duration was met for each supra-threshold event; and (3) The temporal distance between two successive supra-threshold peaks was less than 200 ms (*Davidson et al., 2009*; *Wu and Foster, 2014*). The remainder of detected SWRs was considered as single/isolated events. $62 \pm 0.03\%$ (mean + SEM; n = 14 mice) of all detected bundled ripples in our data consist of two ripples.

### MUA calculation

MUA signal was calculated from hippocampal and neocortical LFP signals using a similar method reported before (*Belitski et al., 2008*). Briefly, the LFP signals were filtered above 300 Hz, rectified and smoothed with a rectangular window with the length of ~ 3 ms for the hippocampal and 100 ms for the neocortical signals. The resultant signals are called HPC and RSC MUA, respectively, in this work.

### Detection of neocortical up-/down-states

A thresholding procedure (*Battaglia et al., 2004*) similar to what used for the SWRs detection was employed to detect the neocortical up-/down states. Briefly, two thresholds for amplitude and duration were applied on the neocortical MUA signal. The events (1) whose amplitudes were greater (smaller) and (2) whose durations were longer than the corresponding thresholds were considered as up-states (down-states). Different thresholds were used for detection of up- and down-states.

### Detection of neocortical peak activation measured by optical signal

A very similar approach to what used for the detection of neocortical up-/down-states was employed to detect neocortical peak activations measured by optical signal. Briefly, the voltage or iGluSnFR $\Delta F/F_0$ captured from each neocortical region was thresholded for both amplitude and duration. The

events with amplitudes larger and durations longer than the corresponding thresholds were detected as peak activations in that region. Different thresholds were used for different neocortical regions.

### Preprocessing of voltage and iGluSnFR imaging data

The optical voltage and glutamate signals were first denoised by applying singular-value decomposition and taking the components with the greatest associated singular values (*Mitra and Pesaran, 1999*). Then, for each pixel in the imaging window, a baseline ($F_0$) of the optical voltage and iGluSnFR signals (F) was calculated. Baseline calculation was accomplished using the *locdetrend* function in the Choronux toolbox (http://chronux.org/; *Mitra and Bokil, 2007*) to fit a piecewise linear curve to the pixel time series using the local regression method. The calculated baseline signal was then subtracted from the raw signal, and the difference signal was divided by the baseline values at each time point ($\Delta F/F_0$). Because most of the optical signal power is concentrated in lower frequencies (*Mohajerani et al., 2013*), a band pass (0.1–6 Hz) FIR filter was applied on the $\Delta F/F_0$ signal. In this paper, optical signals were expressed as a percentage change relative to the baseline optical signal responses ($\Delta F/F_0 \times 100\%$).

### Optical flow analysis

To estimate the propagation direction of peri-SWR neocortical activity, we randomly selected around twenty peri-SWR stacks of imaging frames, and averaged them. The averaged stack was used to calculate propagation direction of neocortical waves at the time of SWRs. Propagation direction was calculated using the Horn-Schunck method implemented in Matlab-based Optical-Flow Analysis Toolbox developed by *Afrashteh et al. (2017)*. This toolbox can be used to quantify the spatiotemporal dynamics of mesoscale brain activity such as propagation direction, location of sources and sinks as well as the trajectories and temporal velocities of flow of activity dynamics. To generate a circular distribution of propagation direction, we repeated the above process for 1000 iterations.

### z-scoring peri-SWR neocortical activity

The inter-SWR intervals were randomly permuted to generate new timestamps. Then, for each random time point, the surrounding imaging frames, capturing one second before to one second after, were temporally aligned. The mean and standard deviation of the corresponding frames were calculated across all the randomly generated time points. In doing so, we generated a mean and a standard deviation stack of frames. We used these two stacks to z-score all the individual peri-SWR stacks of neocortical activity.

### Statistical tests

All statistical tests for linear data were performed using MATLAB built-in functions. One-sided paired t-test and Wilcoxon signed-rank tests were used for linear data. For circular data, Kuiper two-sample test, implemented in CircStat toolbox (*Berens, 2009*), was used.

### Data exclusion criterion

For some group data comparisons in figure supplements, outlier data points, defined as points outside 95% of the distribution relative to the median value, were excluded in some groups. Whenever the number of animals is less than 14 (n < 14), which is the case only in supplementary figures, such exclusion has been performed.

## Acknowledgements

This work was supported by Natural Sciences and Engineering Research Council of Canada (grant no 40352 and 1631465 to MHM and BLM respectively), Alberta Innovates (MHM and BLM), Alberta Prion Research Institute (grant no. 43568 to MHM), and Canadian Institute for Health Research (grant no 390930 and 156040 to MHM and BLM MHM respectively), National Science Foundation (MHM, BLM), and USA Defense Advanced Research Projects Agency (grant no HR0011-18-2-0021 to BLM). The authors also thank Di Shao and Behroo Mirza Agha for animal breeding, J Sun for surgical

assistance and Masami Tatsuno for their suggestions for data analysis. The authors thank Hongkoi Zeng from the Allen Institute for Brain Science for providing the Emx-cre, Camk2a-tTa, and Ai85 mice as a gift.

## Additional information

### Funding

| Funder | Grant reference number | Author |
|---|---|---|
| Natural Sciences and Engineering Research Council of Canada | 40352 | Majid H Mohajerani |
| Natural Sciences and Engineering Research Council of Canada | 1631465 | Bruce L McNaughton |
| Alberta Innovates - Health Solutions | | Majid H Mohajerani |
| Canadian Institutes of Health Research | 390930 | Majid H Mohajerani |
| Canadian Institutes of Health Research | 156040 | Bruce L McNaughton |
| Defense Advanced Research Projects Agency | HR0011-18-2-0021 | Bruce L McNaughton |
| Alberta Prion Research Institute | 43568 | Majid H Mohajerani |
| National Science Foundation | | Majid H Mohajerani Bruce L McNaughton |

The funders had no role in study design, data collection and interpretation, or the decision to submit the work for publication.

### Author contributions

J Karimi Abadchi, Conceptualization, Data curation, Software, Formal analysis, Validation, Investigation, Visualization, Methodology, Writing - original draft, Writing - review and editing; Mojtaba Nazari-Ahangarkolaee, Edgar Bermudez-Contreras, Investigation, Methodology; Sandra Gattas, Formal analysis, Writing - original draft, Writing - review and editing; Artur Luczak, Formal analysis, Validation; Bruce L McNaughton, Conceptualization, Supervision, Funding acquisition, Methodology, Writing - review and editing; Majid H Mohajerani, Conceptualization, Resources, Supervision, Funding acquisition, Methodology, Writing - review and editing

### Author ORCIDs

J Karimi Abadchi (iD) https://orcid.org/0000-0003-4175-7598
Sandra Gattas (iD) https://orcid.org/0000-0003-1608-1469
Edgar Bermudez-Contreras (iD) https://orcid.org/0000-0002-4937-1780
Majid H Mohajerani (iD) https://orcid.org/0000-0003-0964-2977

### Ethics

Animal experimentation: The animal housing, handling, and surgery protocols (#1812) were approved by the University of Lethbridge Animal Care Committee and were in accordance with guidelines set forth by the Canadian Council for Animal Care.

### Decision letter and Author response

Decision letter https://doi.org/10.7554/eLife.51972.sa1
Author response https://doi.org/10.7554/eLife.51972.sa2

# Additional files

## Supplementary files
• Transparent reporting form

## Data availability

All data analyzed and used to produce the main findings of this study have been deposited on Dryad. Source data files have been provided for Figures 2, 3, 5, and 7.

The following dataset was generated:

| Author(s) | Year | Dataset title | Dataset URL | Database and Identifier |
|---|---|---|---|---|
| KarimiAbadchi J, Nazari-Agangarko-laee M, Gattas S, Bermudez-Con-treras E, McNaughton BL, Mohajerani MH | 2019 | Spatiotemporal patterns of neocortical activity around hippocampal sharp-wave ripples | https://doi.org/10.5061/dryad.qnk98sfbb | Dryad Digital Repository, 10.5061/dryad.qnk98sfbb |

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
