## [Decision Letter]

Thank you for submitting your article "Spatiotemporal patterns of neocortical activity around hippocampal sharp-wave ripples" for consideration by *eLife*. Your article has been reviewed by three peer reviewers, one of whom is a member of our Board of Reviewing Editors, and the evaluation has been overseen by Laura Colgin as the Senior Editor.

The reviewers have discussed the reviews with one another, and the Reviewing Editor has drafted this decision to help you prepare a revised submission.

This paper combines wide field optical recording techniques with electrophysiology to study cortico-hippocampal interactions surrounding the hippocampal sharp wave-ripple (SWR). The timing of activation and deactivation of cortex appears continuously distributed with a bias for cortical activation before SWR. Within the cortex, activation appears to propagate from medial to lateral/sensory regions. Interestingly, the propensity to have SWRs within a short window ("bundles") is correlated with the hippocampus activation before cortical activation.

The reviewers agree that this work uses a novel and potentially powerful experimental approach to address important questions of hippocampal-cortical communication. The manuscript is clearly written and reflects a substantial scientific work. However, the reviewers also raise several substantial concerns about the quantitative statistics used and interpretation of the observed results, which must be addressed. The reviewers also suggest new analyses that will significantly increase the relevance and appeal of the present study.

Essential revisions:

1) The findings of hippocampal MUA activity before and after SWRs correlating with neocortical activation is very sensitive to the accuracy of SWR detection. If SWR power is elevated before/after a detected SWR but doesn't pass the detection threshold, the observed MUA may simply reflect hippocampal SWR-related firing. In this case cortical activity coordinated in time would be expected. This could influence the results of both Figure 5 and Figure 6. The authors should analyze their data to show that this is not the case or reinterpret the findings.

2) There are a number of cases where it seems like inappropriate statistical tests were performed in a way that could strongly influence the results. For example, in Figures 2ciii, 2dii, 3cii. It seems like one-sided paired t-tests were conducted between a chosen subset of the groups. This is faulty for a number of independent reasons: pairwise instead of a group test, selective choosing of the groups without a-priori strong reasons to do so, inexplicable use of one-tail test even to make claims explicitly not permitted by the use of one-sided tests (e.g. claiming that "visual (activation is) followed by the medial (activation)" using medial vs visual one-sided test showing no statistical significance), no correction for multiple comparisons, small sample size for a t-test. Repeated-Measures ANOVA or other appropriate tests should be used throughout. Later in the paper the authors switch to non-parametric tests, but it is unclear why.

3) The idea of a loop-like interaction between neocortex and hippocampus during SWS ripples is not new. For example, it has been similarly expressed by Rothschild et al., 2017 and 2018, and recently summarized by Klinzing et al., 2019. This literature needs to be more elaborately integrated. It would be of great interest here to actually demonstrate this loop in the data (i.e., to see if cortical activation that triggers ripples also triggers delayed cortical activations). Ideally, this analysis should be separate for naturally sleeping animals (which optimally were exposed to a novel environment before imaging) and anesthetized animals.

4) The authors could greatly enhance the impact of their work by adding an analysis of the neocortical slow oscillation (SO). Ripples are well known to more likely occur in the down-to-up transition of the SO, and the SO down-state is often considered a global frame resetting activity also in hippocampal networks. Indeed, the authors on several occasions discuss the possibility that the decreases and increases in MUA they observe around ripples represent down and up states of the SO. Why don't they add the respective data and analyses?

5) A key finding in this manuscript is that changes in neocortical activity often precede hippocampal SWRs and the potential interpretation that this may support cortical biasing of hippocampal reactivation. While the data seems to support this proposal, the authors downplay important previous studies that have described such findings (in some cases with neural population data with single-cell resolution) and models. Sirota et al., 2003, reported somatosensory cortical firing preceded SWRs and suggested a cortical influence on hippocampal reactivation; Wang and Ikemoto, 2016, found a similar pattern in ACC; Rothschild et al., 2017, identified a cortical-hippocampal-cortical loop of communication around SWRs and proposed a related model (Rothschild, 2018); Recent human studies found a similar pattern of communication (Norman et al., 2019, Viet-Ngo et al., 2019, Helfrich et al., 2019). These previous reports of this phenomenon should be highlighted.

6) The functional significance of a key result in the paper- the enhanced SWR-related activity in RSC as compared to other cortical regions- is unclear. On the one hand, we know that RSC receives direct input from hippocampus whereas other regions do not. On the other hand, degree of fluorescence does not necessarily tell us much about the functional role of the communication. So what do we learn from this finding beyond a reflection of known anatomy?

7) Given that the important physiological signatures differ across the groups (Figure 2—figure supplement 1), the reviewers questioned the appropriateness of grouping highly different experimental groups/conditions, such as in VSD/glutamate imaging and natural sleep/anesthesia. Do the authors have more data that can enable them to separately analyze these groups? If not, they must provide appropriate justification for the grouping and demonstrate that the different groups pooled together did not perform differently from each other in specific analyses.

8) Figure 1—figure supplement 1 shows recording electrode at the CA1/subiculum boundary, and not unambiguously in CA1 as claimed by the authors. Authors need to show the distribution of all LFP recording electrodes. On a related note, was there any functional mapping performed to demarcate the boundaries between different cortical regions, or are the boundaries based on the atlas?

9) A number of methodological restrictions need to be clearly discussed. First, wide field imaging did not cover prefrontal cortex. In particular, medial prefrontal cortex is an area most strongly connected to the hippocampus and there are a number of previous studies showing particular temporal "peri-ripple" relationships of activity between mPFC and hippocampus. The retrosplenial cortex being the more important hub for sleep dependent consolidation (than mPFC) very much fits with some of the human fMRI literature (e.g., Darsaud et al., 2011, J Cogn Neurosci), although such conclusions unfortunately cannot be made based on the present data. While cortical activity is assessed over rather large areas, hippocampal ripples are recorded from just one electrode site. Considering the local (and travelling) nature of ripples, the temporal relationship of ripples to neocortical activity might essentially depend on where in the hippocampus they are generated/recorded. A related conceptual issue (to be discussed) is that local memory replay mainly occurs during the ripple, and not before or after.

---

## [Author Response]

Essential revisions:1) The findings of hippocampal MUA activity before and after SWRs correlating with neocortical activation is very sensitive to the accuracy of SWR detection. If SWR power is elevated before/after a detected SWR but doesn't pass the detection threshold, the observed MUA may simply reflect hippocampal SWR-related firing. In this case cortical activity coordinated in time would be expected. This could influence the results of both Figure 5 and Figure 6. The authors should analyze their data to show that this is not the case or reinterpret the findings.

We agree that the elevation of hippocampal MUA around detected SWRs could sometimes be due to the presence of undetected SWRs, mainly because there is a significant correlation between HPC MUA and HPC ripple power signals. However, we argue that it does not influence the results reported in Figures 5 and 6. We have made the following cartoons to explain our arguments. For any detected SWR, one of the following four cases shown in Author response image 1 could occur.

For SWRs in case iv, the detection accuracy is not relevant since there is no other undetected SWR around them. For cases i-iii, the accuracy of SWR detection could, in theory, influence the results presented in Figures 5 and 6. To illustrate how the accuracy of ripple detection could affect the hippocampal-neocortical temporal relationship, we considered the three following possibilities (a-c) for the timing of neocortical activation relative to the detected/undetected SWRs.

In (b), detection of the first or second SWR (in temporal order) does not affect the matching between skewness of HPC MUA and RSC signals because (1) If the first SWR was detected, both HPC MUA and as shown in Author response image 2 RSC signals would be skewed to the right. (2) If the second SWR was detected, both HPC MUA and RSC signals would be skewed to the left. Thus, the accuracy of SWR detection does not affect the results for SWRs in cases of i-iii.

**Author response image 2. respfig2:** 

In (a) and (c), the detection of first or second SWR influences the matching between skewness of HPC MUA and RSC signals. In (a), if the first SWR was detected, HPC MUA and RSC signals would be skewed to the right and left, respectively. However, if the second SWR was detected, both HPC MUA and RSC signals would be skewed to the left. A similar discussion can be made for (c) for which only the detection of the second SWR leads to a mismatch between skewness of HPC MUA and RSC signals.

Therefore, for only 2 out of 18 scenarios (3 for i-iii; 3 for a-c; and 2 for detection of first or second SWR; 3*3*2 = 18), the accuracy of SWR detection could yield a mismatch between skewness of HPC MUA and RSC signals. These two scenarios (combinations ii-a and iii-c) are least expected to occur. This is because in a bundled ripple the RSC signal peaks between the first and second SWRs (between 0 and 0.2 s; see Materials and methods and Figure 7—figure supplement 1 panel Ci). Therefore the accuracy of SWR detection minimally influences the matching between skewness of HPC MUA and RSC signals.

To further control for the effect of undetected SWRs in the results shown in Figures 5 and 6, we also reanalyzed our data and made new figures for the relevant panels in Figures 5 and 6. The newly generated figures/panels are labelled similarly to their corresponding figures/panels in the manuscript. Here we focused on isolated/single SWRs (case iv) to reduce the probability of the presence of undetected SWRs, and its possible impact on the temporal order of hippocampal-neocortical interactions. Isolated/single SWRs were detected by applying a second thresholding procedure (Materials and method subsection “Detection of neocortical up-/down-states”) to the already detected SWRs used for Figures 1-6. Hence, the probability of the existence of undetected SWRs around isolated/single SWRs is low. The results of this new analysis match those reported in Figures 5 and 6.

2) There are a number of cases where it seems like inappropriate statistical tests were performed in a way that could strongly influence the results.

We appreciate the reviewers’ comment and we have now applied repeated-measures ANOVA tests wherever was appropriate. We also compared all possible pairs of conditions, corrected for multiple comparisons, and revised the manuscript and figures accordingly.

For example, in Figures 2ciii, 2dii, 3cii. It seems like one-sided paired t-tests were conducted between a chosen subset of the groups. This is faulty for a number of independent reasons: pairwise instead of a group test, selective choosing of the groups without a-priori strong reasons to do so, inexplicable use of one-tail test even to make claims explicitly not permitted by the use of one-sided tests (e.g. that "visual (activation is) followed by the medial (activation)" using medial vs visual one-sided test showing no statistical significance), no correction for multiple comparisons, small sample size for a t-test. Repeated-Measures ANOVA or other appropriate tests should be used throughout.

For Figures 2Ciii, 2Dii, 3Cii, we applied the repeated-measures ANOVA with Greenhouse-Geisser correction for sphericity assumption, followed by multiple comparison tests for all possible pairs of subnetworks. This hypothesis testing did not alter the statistical differences initially reported except for two comparisons in Figure 2Ciii

Later in the paper the authors switch to non-parametric tests, but it is unclear why.

Non-parametric statistical tests were used for Figures 5 (panels Aii and Bv) and 7 (panels Biii*-*iv, C, and D),because the assumption of normality was violated. This is because the distributions are skewed and are not symmetric with respect to their medians (see boxplots in Figures 5 and 7). Although not used in the manuscript, the application of paired t-tests to all the comparisons also resulted in *p*-values very close to the ones reported in the manuscript.

3) The idea of a loop-like interaction between neocortex and hippocampus during SWS ripples is not new. For example, it has been similarly expressed by Rothschild et al., 2017 and 2018, and recently summarized by Klinzing et al., 2019. This literature needs to be more elaborately integrated.

We agree that this idea is not new and has been previously proposed by others. The abovementioned literature is now incorporated into the manuscript.

It would be of great interest here to actually demonstrate this loop in the data (i.e., to see if cortical activation that triggers ripples also triggers delayed cortical activations). Ideally, this analysis should be separate for naturally sleeping animals (which optimally were exposed to a novel environment before imaging) and anesthetized animals.

We agree that showing the neocortical-hippocampal-neocortical loop in our current data would be of great interest. One way of detecting such a potential loop is to check if the neocortical activations that precede SWRs (RSC QR1 in Figure 5Bii) also trigger delayed neocortical activations. To check for the presence of such delayed neocortical activations, we calculated the mean of the peri-SWR RSC activity for SWRs in the RSC QR1 (blue curve shown in Figure 5Bii) for each animal and then averaged the resultant signals across animals. The results of this analysis are shown in Figure 5—figure supplement 3. The averaging was performed separately for data recorded under sleep and urethane anesthesia.

As is evident from this figure, there is no indication of the existence of a delayed activation and, therefore, possibly no neocortical-hippocampus-neocortical loop. The lack of neocortical-hippocampal-neocortical loop in our data could be explained by the extent of animal experience prior to data collection. Rothschild et al., 2017, trained rats for at least 12 days in a spatial task, while the mice used in our experiment were naïve. Although unlikely, the lower temporal resolution of glutamate and voltage imaging, compared with that of single-unit activity presented in Rothschild et al., 2017, may also explain why we did not observe a neocortical-hippocampal-neocortical loop in our data.

4) The authors could greatly enhance the impact of their work by adding an analysis of the neocortical slow oscillation (SO).

We agree with the reviewers thatputting the peri-SWR neocortical activity in the context of neocortical slow oscillations (SOs) can strengthen the connection of our results with those of the numerous research papers focused on studying hippocampal-neocortical interactions. The results presented in Figure 3 possibly reflect the relationship between SWRs and SOs. In Figure 3, the pattern of mean hippocampal ripple power around neocortical peak activations, recorded by optical imaging, is reported. Since there is a noticeable correspondence between neocortical activity measured with LFP and optical imaging (Figure 1Cii), we expect that neocortical activations and deactivations, measured with optical imaging, are likely the reflection of neocortical up- and down-states, respectively.

Ripples are well known to more likely occur in the down-to-up transition of the SO, and the SO down-state is often considered a global frame resetting activity also in hippocampal networks. Indeed, the authors on several occasions discuss the possibility that the decreases and increases in MUA they observe around ripples represent down and up states of the SO. Why don't they add the respective data and analyses?

SOs and up-/down-states were originally characterized using neocortical electrophysiological recordings. Thus, we used the electrophysiological signals recorded from the neocortex (RSC) in a number of animals (n = 4) to investigate the relationship between SOs and SWRs. We have now made a new figure (Figure 3—figure supplement 2) that shows the fluctuations of hippocampal ripple power around neocortical up/down-states detected using neocortical MUA.

As expected, the ripple power decreased before and increased after the onset of neocortical up-states. Moreover, the ripple power increased before and decreased after the onset of neocortical down-states. Therefore, the results presented in Figure 3 are reflections of the relationship between SWRs and SOs. We have now added new sentences to the manuscript to state the results presented in Figure 3—figure supplement 2. These sentences read:

“Moreover, we investigated the relationship between neocortical slow oscillation and hippocampal SWRs in animals for which neocortical electrophysiological signals were recorded (Figure 3—figure supplement 2). We found that hippocampal ripple power is suppressed (elevated) before (after) the onset of neocortical up-states. Similarly, hippocampal ripple power is elevated (suppressed) before (after) the onset of neocortical down-states. Therefore, the results obtained using optical imaging and neocortical electrophysiology were compatible.”

“Consistent with previous reports, we also found a strong modulation of hippocampal ripple power around the onset of neocortical up-/down-states. Moreover, we extended the previous findings by providing the spatial distribution of correspondence between neocortical up-states and hippocampal ripple power.”

5) A key finding in this manuscript is that changes in neocortical activity often precede hippocampal SWRs and the potential interpretation that this may support cortical biasing of hippocampal reactivation. While the data seems to support this proposal, the authors downplay important previous studies that have described such findings (in some cases with neural population data with single-cell resolution) and models. Sirota et al., 2003, reported somatosensory cortical firing preceded SWRs and suggested a cortical influence on hippocampal reactivation; Wang and Ikemoto, 2016, found a similar pattern in ACC; Rothschild et al., 2017, identified a cortical-hippocampal-cortical loop of communication around SWRs and proposed a related model (Rothschild, 2018); Recent human studies found a similar pattern of communication (Norman et al., 2019, Viet-Ngo et al., 2019, Helfrich et al., 2019). These previous reports of this phenomenon should be highlighted.

The abovementioned studies have been incorporated into the manuscript.

6) The functional significance of a key result in the paper- the enhanced SWR-related activity in RSC as compared to other cortical regions- is unclear.

We agree with the reviewers that with our experimental design, we cannot determine the functional significance of enhanced SWR-related activity in RSC relative to other neocortical regions. While we think it is an important question for future studies, it is difficult to completely address this issue within the confines of the current manuscript. We are planning to conduct new sets of experiments in which we will examine the sufficiency and/or necessity of this enhanced activity for the consolidation of recently acquired memories.

On the one hand, we know that RSC receives direct input from hippocampus whereas other regions do not. On the other hand, degree of fluorescence does not necessarily tell us much about the functional role of the communication. So what do we learn from this finding beyond a reflection of known anatomy?

Although there is a strong relationship between structure and function in the brain, the presence of structural projections between two brain structures cannot fully explain the dynamics of interactions. Here are some examples:

1) Given that the projections between the hippocampus (HPC) and RSC are mainly unidirectional (from HPC to RSC and not the other way around), one may predict that hippocampal SWRs would activate RSC. Instead, we found that RSC, on average, tends to activate and peak before hippocampal SWRs. Therefore, the RSC activation prior to HPC SWRs cannot easily be explained by the pattern of anatomical connections between the two structures.

2) In addition to RSC, we found that secondary motor, posterior parietal, visual, and auditory cortices were also modulated around SWRs, each with a distinct time course (Figure 2—figure supplement 3C and Figure 4Aii). This finding cannot necessarily be predicted by anatomy since HPC mainly projects to RSC and not to other neocortical regions.

3) The midline areas including (not restricted to) RSC tend to activate earlier than more lateral neocortical regions (e.g., somatosensory cortices) around SWRs. Moreover, neocortical activation waves tend to propagate in the mediolateral direction. This also cannot be easily explained based on known structural connectivity between HPC and RSC.

4) The majority of studies on peri-SWR hippocampal-neocortical interactions in rodents have been focused on recording from the dorsal CA1 field of the hippocampus and the medial prefrontal cortex (mPFC), while the primary projections from the HPC to mPFC emanate from the ventral portion of HPC.

Given the absence of direct structural connections between the dorsal hippocampus and mPFC, one may predict that there would be no functional interactions between the two structures. However, despite the lack of direct structural connections, important discoveries on the functional significance of interactions between dorsal HPC and mPFC in the context of learning and memory have been found.

7) Given that the important physiological signatures differ across the groups (Figure 2—figure supplement 1), the reviewers questioned the appropriateness of grouping highly different experimental groups/conditions, such as in VSD/glutamate imaging and natural sleep/anesthesia. Do the authors have more data that can enable them to separately analyze these groups? If not, they must provide appropriate justification for the grouping and demonstrate that the different groups pooled together did not perform differently from each other in specific analyses.

We do not have enough data to analyze each group separately, nor did we plan to compare these conditions. The main reason for not collecting more data under natural sleep was the low success rate of having animals sleeping under head fixation, especially with a slightly tilted head. Tilting the head was done to include temporal cortices (e.g., the auditory cortex) within our imaging window. Thus, we used a model of sleep (urethane anesthesia) to increase our sample size. We had justified the pooling of sleep with anesthesia data in the first sub-section of the Results section, and in Figure 2—figure supplement 3A-D. To further justify the pooling, we compared all the reported measures in this study between VSD and iGluSnFR imaging, as well as between sleep and urethane conditions. We have now made additional supplementary figures (Figure 2—figure supplement 3E-F; Figure 3—figure supplement 1; Figure 4—figure supplement 2; Figure 5—figure supplement 2; Figure 6—figure supplement 2; Figure 2—figure supplement 7) that include the results of the abovementioned comparisons across all conditions/groups. Aside from a handful of exceptions (see Figure 2—figure supplement 3Eii; Figure 3—figure supplement 1ii; Figure 5—figure supplement 2Bi; Figure 2—figure supplement 7Ai and Cii), the rest of the comparisons did not yield any significant differences between these groups/conditions. We have now added a new sentence to emphasize this point in the manuscript. This sentence reads:

“We confirmed that the quantities, reported in the following sections, were not different across urethane anesthesia versus sleep and VSD versus iGluSnFR imaging conditions (Figure 2—figure supplement 3; Figure 3—figure supplement 1; Figure 4—figure supplement 2; Figure 5—figure supplement 2; Figure 6—figure supplement 2; Figure 7—figure supplement 2).”

We would also like to emphasize that we replicated some of the imaging results using multi-site electrophysiological recordings in unrestrained, naturally-sleeping animals (see Figure 2—figure supplement 2 and 5). These results were already included in our first submission.

8) Figure 1—figure supplement 1 shows recording electrode at the CA1/subiculum boundary, and not unambiguously in CA1 as claimed by the authors. Authors need to show the distribution of all LFP recording electrodes.

We have now included the location of LFP electrode tips in 14 mouse brains in Figure 1—figure supplement 1 panel iv. The electrode insertion procedure and spatial coordinates used to fix the electrodes were the same for all 14 animals. The distribution of these 14 electrode tips is represented in the following figure. The location of tips is categorized into 4 groups, each of which is colored differently. The number of animals in each category is reported on the figure.

The majority (12 out of 14) of the electrodes were either in the CA1 (green and cyan) or at the border of CA1 and subiculum (red). As ripple events are synchronous between CA1 and ipsilateral subiculum with high probability (Chrobak and Buzsáki, 1996), no change in our findings is anticipated for animals for which the electrode tips were located at the border of CA1 and subiculum.

On a related note, was there any functional mapping performed to demarcate the boundaries between different cortical regions, or are the boundaries based on the atlas?

The centers of sensory regions were found by periphery stimulation, as is explained in the Materials and methodssection. The boundaries of the regions were based on the Paxinos and Franklin mouse brain atlas (Paxinos, G. and Franklin, K.B.J. The Mouse Brain in Stereotaxic Coordinates, Elsevier Academic Press, Amsterdam, Boston, 2004).

9) A number of methodological restrictions need to be clearly discussed. First, wide field imaging did not cover prefrontal cortex. In particular, medial prefrontal cortex is an area most strongly connected to the hippocampus and there are a number of previous studies showing particular temporal "peri-ripple" relationships of activity between mPFC and hippocampus.

We agree with the reviewers on their note about the absence of mPFC in our imaging window. This is now emphasized in the revised manuscript (subsection “Neocortex tends to activate sequentially from medial to more lateral regions around SWRs”).

The retrosplenial cortex being the more important hub for sleep dependent consolidation (than mPFC) very much fits with some of the human fMRI literature (e.g., Darsaud et al., 2011, J Cogn Neurosci), although such conclusions unfortunately cannot be made based on the present data.

We agree with the reviewers’ point that no conclusion about the comparative role of RSC and mPFC in memory consolidation can be made based on our data. Indeed, no comment on this point was made in the manuscript, nor was our goal to make this comparison.

While cortical activity is assessed over rather large areas, hippocampal ripples are recorded from just one electrode site. Considering the local (and travelling) nature of ripples, the temporal relationship of ripples to neocortical activity might essentially depend on where in the hippocampus they are generated/recorded. A related conceptual issue (to be discussed) is that local memory replay mainly occurs during the ripple, and not before or after.

We agree with the reviewers’ comment regarding the temporal relationship between propagating SWRs and neocortical activity. We revised the manuscript and included the following paragraph in the Discussion section.

“We should emphasize that the propagating nature of SWRs (Patel et al., 2013) does not influence the results reported in the current study. The majority of SWRs propagate less than 3 mm across the septal intermediate or intermediate-temporal portion of the hippocampus with a minimum speed of 0.33 mm/ms (Patel et al., 2013). Considering the lengths of mouse hippocampus septal-intermediate and intermediate temporal portions (~3.25mm)(Milior et al., 2016), it would take less than 9 ms (equivalent to one cycle of ripples) for SWRs to reach to the electrode tip from their initiation zones. Therefore, changing the location of the electrode tip in the septal-intermediate portion of the hippocampus would have a minimal effect on the results reported in this study. However, our findings could have been different had we recorded from the intermediate-temporal portion of HPC since SWRs in dorsal and ventral portions of the hippocampus often do not co-occur (Patel et al., 2013).”